# CONFORMAL BOUNDS ON FULL-REFERENCE IMAGE QUALITY FOR IMAGING INVERSE PROBLEMS

## ABSTRACT

In imaging inverse problems, we would like to know how close the recovered image is to the true image in terms of full-reference image quality (FRIQ) metrics like PSNR, SSIM, LPIPS, etc. This is especially important in safety-critical applications like medical imaging, where knowing that, say, the SSIM was poor could potentially avoid a costly misdiagnosis. But since we don't know the true image, computing FRIQ is non-trivial. In this work, we combine conformal prediction with approximate posterior sampling to construct bounds on FRIQ that are guaranteed to hold up to a user-specified error probability. We demonstrate our approach on image denoising and accelerated magnetic resonance imaging (MRI) problems.

## 1 INTRODUCTION

In imaging inverse problems, one aims to recover a true image $x_0$ from noisy/distorted/incomplete measurements $y_0 = \mathcal{A}(x_0)$ (Arridge et al., 2019). Denoising, deblurring, inpainting, super-resolution, limited-angle computed tomography, and accelerated magnetic resonance imaging (MRI) are examples of linear inverse problems, while phase-retrieval, de-quantization, low-light imaging, and image-to-image translation are examples of non-linear inverse problems. Such problems are ill-posed, in that many hypotheses of $x_0$ are consistent with both the measurements $y_0$ and prior knowledge about $x_0$. To complicate matters, different recovery methods are biased towards different plausible image hypotheses, leading to important differences in reconstruction quality. For example, modern deep-network approaches can sometimes hallucinate (Belthangady & Royer, 2019; Hoffman et al., 2021; Muckley et al., 2021; Bhadra et al., 2021; Gottschling et al., 2023), i.e., generate visually pleasing recoveries that differ in important ways from the true image $x_0$. Thus, there is a strong need to quantify the accuracy of a given recovery, especially in safety-critical applications like medical imaging (Banerji et al., 2023).

In image recovery, "accuracy" can be defined in different ways. Classical metrics like mean-squared error (MSE), or its scaled counterpart peak signal-to-noise ratio (PSNR), are convenient for theoretical analysis but do not always correlate well with human perceptions of image quality. This fact inspired the field of full-reference image-quality (FRIQ) assessment (Lin & Kuo, 2011; Wang, 2011), which led to the well-known Structural Similarity Index Measure (SSIM) (Wang et al., 2004b) that is still popular today. However, progress continues to be made. Most recent methods leverage the internal features of deep neural networks, which are said to mimic the processing architecture of the human visual cortex (Yamins & DiCarlo, 2016; Lindsay, 2021). A popular example of the latter is Learned Perceptual Image Patch Similarity (LPIPS) (Zhang et al., 2018). In the end, though, the best choice of metric may depend on the application. For example, in magnetic resonance imaging (MRI), the goal is to provide the radiologist with an image recovery that leads to an accurate diagnosis. A recent clinical MRI study (Kastryulin et al., 2023) found that, among 35 tested metrics, Deep Image Structure and Texture Similarity (DISTS) (Ding et al., 2020a) correlated best with radiologists' perceptions.

In this work, our goal is to provide rigorous bounds on the FRIQ $m(\widehat{x}_0, x_0)$ of a recovery $\widehat{x}_0 = h(y_0)$ relative to the true image $x_0$. Here, $h(\cdot)$ is an arbitrary image-recovery scheme and $m(\cdot, \cdot)$ is an arbitrary FRIQ metric. The key challenge is that $x_0$ is unknown. To our knowledge, there exists no prior work on providing FRIQ guarantees in image recovery. Our contributions are as follows.

1. We propose a framework to bound the FRIQ $m(\widehat{x}_0, x_0)$ of a recovered image $\widehat{x}_0$ without access to the true image $x_0$. Our framework uses conformal prediction (Vovk et al., 2005; Angelopoulos & Bates, 2023) to construct bounds that hold with probability at least $1 - \alpha$ under certain exchangeability assumptions and where $\alpha \in (0, 1)$ is chosen by the user.

2. We show how posterior-sampling-based image recovery can be used to construct conformal bounds that adapt to the measurements $y_0$ and reconstruction $\widehat{x}_0$.

3. We demonstrate our approach on two linear inverse problems: denoising of FFHQ faces (Karras et al., 2019) faces and recovery of fastMRI knee images (Zbontar et al., 2018) from accelerated multicoil measurements.

From the perspective of uncertainty quantification (UQ), one could say that our goal is to bound the uncertainty on FRIQ $m(\widehat{x}_0, x_0)$ that arises due to $x_0$ being unknown. As such, our approach to UQ differs from typical ones in image recovery. There, uncertainty is typically quantified on individual pixels, with the overall result being a pixel-wise uncertainty map. To construct these maps, one could use (approximate) posterior samplers (Durmus et al., 2018; Laumont et al., 2022; Zach et al., 2022; Tonolini et al., 2020; Edupuganti et al., 2021; Adler & Öktem, 2018; Bendel et al., 2023; Ardizzone et al., 2019; Wen et al., 2023a; Jalal et al., 2021; Chung et al., 2023) or Bayesian neural networks (BNNs) (Kendall & Gal, 2017; Xue et al., 2019; Barbano et al., 2021; Ekmekci & Cetin, 2022; Narnhofer et al., 2022), including those based on dropout (Kendall & Gal, 2017), to draw many reconstructions from the distribution of plausible $x_0$ for a given $y_0$ (i.e., the posterior distribution $p_{X_0|Y_0}(\cdot|y_0)$), from which pixel-wise standard-deviations can be estimated. An alternative is to utilize conformal prediction to produce pixel-wise intervals that are guaranteed to contain the true pixel value with high probability (Angelopoulos et al., 2022b; Horwitz & Hoshen, 2022; Teneggi et al., 2023; Kutiel et al., 2023; Narnhofer et al., 2024). Although these uncertainty maps can be visually interesting, they do not quantify uncertainty on multi-pixel structures of interest, such as hallucinations or anatomical features relevant to medical diagnosis (e.g., tumors).

To our knowledge, there exist relatively few works on multi-pixel UQ, and none target FRIQ. For example, Tang & Repetti (2023) use hypothesis testing to infer the presence/absence of a structure-of-interest within the maximum a posteriori (MAP) image recovery, but relies on inpainting to construct the structure-absent hypothesis, which may not be accurate. Sankaranarayanan et al. (2022) use conformal prediction to compute uncertainty intervals on the presence/absence of semantic attributes (e.g., whether a face has a smile, glasses, etc.) but their method requires a "disentangled" generative adversarial network (GAN) that generates image samples given attribute probabilities. Belhasin et al. (2023) compute conformal prediction intervals on the principal components of the posterior covariance matrix. Lastly, given measurements $y_0 = \mathcal{A}(x_0)$ and a downstream imaging task $\mu(\cdot) \in \mathbb{R}$ (e.g., soft-output classification), Wen et al. (2024) compute conformal bounds on the true task output $\mu(x_0)$. While interesting, none of the above works quantify the uncertainty on FRIQ metrics like PSNR, SSIM, LPIPS, DISTS, etc., due to $x_0$ being unknown.

## 2 BACKGROUND

Conformal prediction (CP) (Vovk et al., 2005; Angelopoulos & Bates, 2023) is a powerful framework for computing uncertainty intervals on the output of any black-box predictor. CP makes no assumptions on the distribution of the data, yet provides probabilistic guarantees that the true target lies within the constructed uncertainty interval. In this paper, we focus on the common variant known as split CP (Papadopoulos et al., 2002; Lei et al., 2018).

We now provide a brief background on split CP. Given features $u_0 \in \mathcal{U}$, the goal of CP is to construct a set $\mathcal{C}_\lambda(\widehat{z}_0)$ that contains an unknown target $z_0 \in \mathcal{Z}$ with high probability. Here, $\mathcal{C}_\lambda(\cdot)$ is constructed so that $|\mathcal{C}_\lambda(\widehat{z}_0)|$ is monotonically non-decreasing in $\lambda \in \mathbb{R}$ for any fixed $\widehat{z}_0$, and $\widehat{z}_0 = f(u_0)$ is some prediction from a black-box model $f(\cdot)$. Split CP accomplishes this goal by calibrating $\lambda$ using a dataset of feature and target pairs $\{(u_i, z_i)\}_{i=1}^n$ that has not been used to train $f(\cdot)$. In particular, it first constructs the set $d_{\mathsf{cal}} \triangleq \{(\widehat{z}_i, z_i)\}_{i=1}^n$ using $\widehat{z}_i = f(u_i)$ and then finds a $\widehat{\lambda}(d_{\mathsf{cal}})$ to provide the marginal coverage guarantee (Lei & Wasserman, 2014)

$$\Pr\left\{Z_0 \in \mathcal{C}_{\widehat{\lambda}(D_{\mathsf{cal}})}(\widehat{Z}_0)\right\} \geq 1 - \alpha, \tag{1}$$

where $\alpha$ is a user-chosen error rate. Here and in the sequel, we use capital letters to denote random variables and lower-case letters to denote their realizations. In words, (1) guarantees that the unknown

target $Z_0$ falls within the interval $\mathcal{C}_{\widehat{\lambda}(D_{\mathsf{cal}})}(\widehat{Z}_0)$ with probability at least $1 - \alpha$ when averaged over the randomness in the test data $(Z_0, \widehat{Z}_0)$ and calibration data $D_{\mathsf{cal}}$.

While there are a number of ways to describe CP calibration of $\lambda$, (Vovk et al., 2005; Angelopoulos & Bates, 2023), we will focus on the method from (Angelopoulos et al., 2022a). It starts by defining the empirical miscoverage as

$$\widehat{r}_n(\lambda; d_{\mathsf{cal}}) \triangleq \frac{1}{n} \sum_{i=1}^{n} \mathbb{1}_{z_i \notin \mathcal{C}_\lambda(\widehat{z}_i)}, \tag{2}$$

where $\mathbb{1}_{\{\cdot\}}$ is the indicator function. The empirical miscoverage measures the proportion of targets $z_i$ that land outside of $\mathcal{C}_\lambda(\widehat{z}_i)$ in the calibration set $d_{\mathsf{cal}}$. Note the dependence on $\lambda$, which controls the size of the prediction interval. The calibration procedure then sets $\lambda$ at

$$\widehat{\lambda}(d_{\mathsf{cal}}) = \inf \left\{ \lambda : \widehat{r}_n(\lambda; d_{\mathsf{cal}}) \leq \alpha - \tfrac{1-\alpha}{n} \right\}, \tag{3}$$

which can be found using a simple binary search. Intuitively, the $\lambda$ chosen in (3) yields an empirical miscoverage that is slightly more conservative than the desired $\alpha$ in order to handle the finite size of the calibration set. When $\{(Z_0, \widehat{Z}_0), (Z_1, \widehat{Z}_1), \ldots, (Z_n, \widehat{Z}_n)\}$ are exchangeable (a weaker condition than i.i.d.), (3) ensures that (1) holds (Angelopoulos et al., 2022a). See the overviews (Angelopoulos & Bates, 2023; Vovk et al., 2005) for more details on conformal prediction.

## 3 PROPOSED APPROACH

Consider an imaging inverse problem, where we observe incomplete and/or noisy measurements $y_0 = \mathcal{A}(x_0)$ of a true image $x_0$. Suppose that $\widehat{x}_0 = h(y_0)$ is a reconstruction of $x_0$ provided by some image recovery method $h(\cdot)$ and that $z_0 = m(\widehat{x}_0, x_0) \in \mathbb{R}$ is some FRIQ metric on $\widehat{x}_0$ with respect to the true $x_0$. We would like to know $z_0$, especially in safety critical applications. For example, if $z_0$ was unacceptable, then perhaps we could use a different recovery method $h(\cdot)$ or collect more measurements $y_0$. But $z_0$ cannot be directly computed because $x_0$ is unknown.

Our key insight is that it's possible to construct a set $\mathcal{C}_\lambda(\widehat{z}_0)$ that is guaranteed to contain the unknown FRIQ $z_0$ with high probability. This can be done using CP, at least when one has access to calibration data $\{(x_i, y_i)\}_{i=1}^n$ of true image and measurement pairs that agrees with the test $(x_0, y_0)$ in the sense that the resulting FRIQ pairs $\{(\widehat{z}_i, z_i)\}_{i=0}^n$ are statistically exchangeable.

Our general approach is as follows. Using $\{(x_i, y_i)\}_{i=1}^n$, we compute the image recovery $\widehat{x}_i = h(y_i)$ and the corresponding true FRIQ $z_i = m(\widehat{x}_i, x_i)$ for each $i = 1, \ldots, n$. Then we construct an estimator $f(\cdot)$ that produces an FRIQ estimate $\widehat{z}_i = f(u_i)$ for some choice of $u_i$. Several choices of $f(\cdot)$ and $u_i$ will be described in the sequel. We then collect the results into the set $d_{\mathsf{cal}} = \{(\widehat{z}_i, z_i)\}_{i=1}^n$ and calibrate the $\lambda$ parameter of the FRIQ prediction interval $\mathcal{C}_\lambda(\widehat{z}_i)$ using CP.

We now describe our choice of prediction interval $\mathcal{C}_\lambda(\cdot)$. In the sequel, we will refer to those metrics $m(\cdot, \cdot)$ for which a higher value indicates better image quality (e.g., PSNR, SSIM) as Higher-Preferred (HP) metrics, and those for which a lower value indicates better image quality (e.g., LPIPS, DISTS) as Lower-Preferred (LP) metrics. We choose to construct the prediction set for the $i$-th sample as

$$\mathcal{C}_\lambda(\widehat{z}_i) = [\beta(\widehat{z}_i, \lambda), \infty) \text{ for HP metrics and } \mathcal{C}_\lambda(\widehat{z}_i) = (-\infty, \beta(\widehat{z}_i, \lambda)] \text{ for LP metrics,} \tag{4}$$

where we choose the lower/upper bound $\beta(\cdot, \cdot)$ as

$$\beta(\widehat{z}_i, \lambda) = \widehat{z}_i - \lambda \text{ for HP metrics and } \beta(\widehat{z}_i, \lambda) = \widehat{z}_i + \lambda \text{ for LP metrics.} \tag{5}$$

By calibrating the bound parameter $\lambda$ as $\widehat{\lambda}(d_{\mathsf{cal}})$ using (3), we obtain the following marginal coverage guarantee for the test sample $(\widehat{Z}_0, Z_0)$:

$$\Pr\left\{ Z_0 \in \mathcal{C}_{\widehat{\lambda}(D_{\mathsf{cal}})}(\widehat{Z}_0) \right\} \geq 1 - \alpha, \tag{6}$$

which holds when $\{(Z_0, \widehat{Z}_0), (Z_1, \widehat{Z}_1), \ldots, (Z_n, \widehat{Z}_n)\}$ are exchangeable (Angelopoulos et al., 2022a). In particular, $\beta(\widehat{Z}_0, \widehat{\lambda}(D_{\mathsf{cal}}))$ lower-bounds the unknown true HP metric value $Z_0$, or upper-bounds the unknown true LP metric value $Z_0$, with probability at least $1 - \alpha$, where $\alpha$ is selected by the

user. A smaller error-rate $\alpha$ will tend to yield a looser bound, but—importantly—the coverage guarantee (6) will hold for any chosen $\alpha \in (0,1)$. In the sequel, we will refer to $\beta(\widehat{z}_0, \widehat{\lambda}(d_{\mathsf{cal}}))$ as the "conformal bound" on $z_0$. Note that the conformal bound can "adapt" to the test measurements $y_0$ and reconstruction $\widehat{x}_0$ through $\widehat{z}_0 = f(u_0)$ for appropriate choices of $f(\cdot)$ and $u_0$.

Below we describe different ways to construct $f(\cdot)$ and $u_0$, which in turn yield conformal bounds with different properties. Appendix D investigates violations of the exchangeability assumption.

### 3.1 A NON-ADAPTIVE BOUND ON RECOVERED-IMAGE FRIQ

As a simple baseline, we start with the choice $f(\cdot) = 0$. In this case, $u_0$ is inconsequential and $\widehat{z}_0 = 0$, and so the conformal bound $\beta(\widehat{z}_0, \widehat{\lambda}(d_{\mathsf{cal}}))$ will depend on the calibration set $d_{\mathsf{cal}}$ but not the test measurements $y_0$ or reconstruction $\widehat{x}_0$. We refer to such bounds as "non-adaptive." As we demonstrate in Sec. 4, non-adaptivity leads to conservative bounds. Still, this non-adaptive bound is valid in the sense of guaranteed marginal coverage (6) under the exchangeability assumption.

### 3.2 INTUITIONS ON CONSTRUCTING ADAPTIVE FRIQ BOUNDS

Our approach to constructing adaptive FRIQ bounds is based on the following probabilistic viewpoint. Conditioned on the observed measurements $y_0$, we can model the unknown FRIQ as $Z_0 = m(\widehat{x}_0, X_0)$ for $\widehat{x}_0 = h(y_0)$ and $X_0 \sim p_{X_0|Y_0}(\cdot|y_0)$. The distribution $p_{X_0|Y_0}(\cdot|y_0)$ is often referred to as the posterior distribution on $X_0$ given the measurements $Y_0 = y_0$.

Let us first consider the ideal and unrealistic case that the $y_0$-conditional FRIQ distribution $p_{Z_0|Y_0}(\cdot|y_0)$ is known. And let's consider the case of HP metrics, noting that similar arguments can be made for LP metrics. If $p_{Z_0|Y_0}(\cdot|y_0)$ was known, then constructing a lower-bound $\beta$ on $Z_0$ that holds with probability $\geq 1 - \alpha$ could be directly accomplished by finding the $\beta \in \mathbb{R}$ that satisfies $\Pr\{Z_0 \geq \beta | Y_0 = y_0\} \geq 1 - \alpha$, which is known as the $\alpha$th quantile of $Z_0 | Y_0 = y_0$.

Now suppose that the distribution of $Z_0 | Y_0 = y_0$ was unknown, but instead one had access to an infinite number of perfect posterior image samples $\{\widetilde{x}_0^{(j)}\}_{j=1}^{\infty}$. By "perfect" we mean that $\widetilde{x}_0^{(j)}$ are independent realizations of $X_0 | Y_0 = y_0$. From them, one could construct posterior FRIQs $\{\widetilde{z}_0^{(j)}\}_{j=1}^{c}$ using $\widetilde{z}_0^{(j)} \triangleq m(\widehat{x}_0, \widetilde{x}_0^{(j)})$. Importantly, $\{z_0, \widetilde{z}_0^{(1)}, \widetilde{z}_0^{(2)}, \widetilde{z}_0^{(3)}, \dots\}$ are i.i.d. realizations of $Z_0 | Y_0 = y_0$. Thus, to construct a lower bound $\beta$ on $Z_0 | Y_0 = y_0$ that holds with probability $1 - \alpha$, one could use the empirical quantile of $\{\widetilde{z}_0^{(j)}\}$, i.e.,

$$\beta = \lim_{c \to \infty} \mathrm{EmpQuant}\left(\alpha, \{\widetilde{z}_0^{(j)}\}_{j=1}^{c}\right), \tag{7}$$

which converges to the $\alpha$th quantile of $Z_0 | Y_0 = y_0$ (Fristedt & Gray, 2013).

In practice, one will not have access to an infinite number of perfect posterior image samples. However, it is not difficult to obtain a *finite* number of *approximate* posterior samples $\{\widetilde{x}_0^{(j)}\}_{j=1}^{c}$. From them, one could estimate the $\alpha$th quantile of $Z_0 | Y_0 = y_0$ and subsequently calibrate that (imperfect) estimate using conformal prediction. Two such strategies are described below.

### 3.3 AN ADAPTIVE BOUND ON RECOVERED-IMAGE FRIQ

Suppose that, for each $i \in \{0, 1, \dots, n\}$, we have access to $c \geq 1$ approximate posterior image samples $\{\widetilde{x}_i^{(j)}\}_{j=1}^{c}$ produced by a black-box posterior image sampler such as those listed in Sec. 1. Guided by the intuitions from Sec. 3.2, we propose the following for HP metrics. For each $i$, we first compute the corresponding approximate posterior FRIQs $\{\widetilde{z}_i^{(j)}\}_{j=1}^{c}$ using $\widetilde{z}_i^{(j)} = m(\widehat{x}_i, \widetilde{x}_i^{(j)})$ and then set $\widehat{z}_i$ at their empirical quantile

$$\widehat{z}_i = \mathrm{EmpQuant}\left(\alpha, \{\widetilde{z}_i^{(j)}\}_{j=1}^{c}\right) = f(u_i) \ \text{ for } \ \begin{cases} f(\cdot) = \mathrm{EmpQuant}(\alpha, \cdot) \\ u_i = [\widetilde{z}_i^{(1)}, \dots, \widetilde{z}_i^{(c)}]^{\top} \in \mathbb{R}^c. \end{cases} \tag{8}$$

We then use $d_{\mathsf{cal}} = \{(\widehat{z}_i, z_i)\}_{i=1}^{n}$ to calibrate the bound parameter $\lambda$ using (3), yielding $\widehat{\lambda}(d_{\mathsf{cal}})$. Finally, we plug this $\lambda$ and $\widehat{z}_0$ into (5) to get $\beta(\widehat{z}_0, \widehat{\lambda}(d_{\mathsf{cal}}))$, which is our conformal bound on the true

Figure 1: Overview of method: Given a recovery $\widehat{x}_0$ of true image $x_0$, approximate posterior samples $\{\widetilde{x}_0^{(j)}\}_{j=1}^c$, and a calibration set $d_{\mathsf{cal}}$, we construct a prediction interval $\mathcal{C}_{\widehat{\lambda}(d_{\mathsf{cal}})}(\widehat{z}_0)$ that is guaranteed to contain the unknown true FRIQ $z_0 = m(\widehat{x}_0, x_0)$ with probability at least $1 - \alpha$.

FRIQ $z_0$. From Sec. 2, we know that this conformal bound satisfies the coverage guarantee (6) under the exchangeability assumption. Furthermore, it adapts to the measurements $y_0$ and reconstruction $\widehat{x}_0$ through their effect on $\widehat{z}_0$ and , unlike the non-adaptive bound from Sec. 3.1.

Recalling Sec. 3.2, one could interpret $\widehat{z}_0$ as a rough estimate of the $\alpha$th quantile of $Z_0 | Y_0 = y_0$ and $\widehat{\lambda}(d_{\mathsf{cal}})$ as an additive correction that accounts for the finite and approximate nature of the posterior image samples $\{\widetilde{x}_0^{(j)}\}_{j=1}^c$ used to construct $\widehat{z}_0$. For LP metrics, we would instead compute the $(1 - \alpha)$-empirical quantile in (8). Figure 1 illustrates the overall methodology.

## 3.4 A LEARNED ADAPTIVE BOUND ON RECOVERED-IMAGE FRIQ

In Sec. 3.2, we reasoned that the $\alpha$th quantile of $Z_0 | Y_0 = y_0$ yields a valid HP FRIQ bound, but we noted that this quantile is not directly observable. Thus, in Sec. 3.3, we used the $\alpha$th empirical quantile of $\{\widetilde{z}_i^{(j)}\}_{j=1}^c$ as a rough estimate "$\widehat{z}_i$" of the desired quantile, after which we used CP to correct this estimate and obtain a valid HP FRIQ bound. However, it is well known from the CP literature that inaccurate base estimators cause loose conformal bounds (Angelopoulos & Bates, 2023). Thus, in this section, we aim to improve our estimate of the $\alpha$th quantile of $Z_0 | Y_0 = y_0$.

Inspired by conformalized quantile regression (Romano et al., 2019), we propose to estimate the $\alpha$th quantile of $Z_0 | Y_0 = y_0$ using

$$\widehat{z}_i = f(u_i; \theta) \ \text{ with } \ u_i = [\widetilde{z}_i^{(1)}, \dots, \widetilde{z}_i^{(c)}]^\top \in \mathbb{R}^c, \tag{9}$$

where $\theta$ are predictor parameters trained using quantile regression (QR) (Koenker & Bassett, 1978). An example $f(\cdot; \theta)$ is given in App. F. In the case of an HP metric, this manifests as

$$\arg\min_\theta \sum_{i=n+1}^{n+n_{\mathsf{train}}} \big(\alpha \max(0, z_i - \widehat{z}_i(\theta)) + (1 - \alpha) \max(0, \widehat{z}_i(\theta) - z_i)\big) + \gamma \rho(\theta), \tag{10}$$

using a training set $d_{\mathsf{train}} = \{(u_i, z_i)\}_{i=n+1}^{n+n_{\mathsf{train}}}$ that is independent of the calibration samples $\{(u_i, z_i)\}_{i=1}^n$ and test sample $(u_0, z_0)$. The first term in (10) is the pinball loss (Koenker & Bassett, 1978), which encourages an $\alpha$-fraction of training samples to violate the HP bound $\widehat{z}_i \leq z_i$. The $\rho(\cdot)$ term in (10) is regularization that avoids overfitting $\theta$ to the training set. The regularization weight $\gamma$ can be tuned using k-fold cross-validation. The $\theta$-dependence of $\widehat{z}_i$ is made explicit in (10).

Once the predictor $f(\cdot; \theta)$ is trained, it can be used to obtain the quantile estimates $\{\widehat{z}_i\}_{i=0}^n$. Then $d_{\mathsf{cal}} \triangleq \{(\widehat{z}_i, z_i)\}_{i=1}^n$ can be used to calibrate the bound parameter $\lambda$ using (3). As before, the resulting conformal bound $\beta(\widehat{z}_0, \widehat{\lambda}(d_{\mathsf{cal}}))$ will enjoy the coverage guarantee (6) under the exchangeability assumption. To handle LP metrics, we would swap $\alpha$ with $1 - \alpha$ in (10). Note that any estimation function $f(\cdot; \theta)$ can be used in (9) and the best choice will vary with the application.

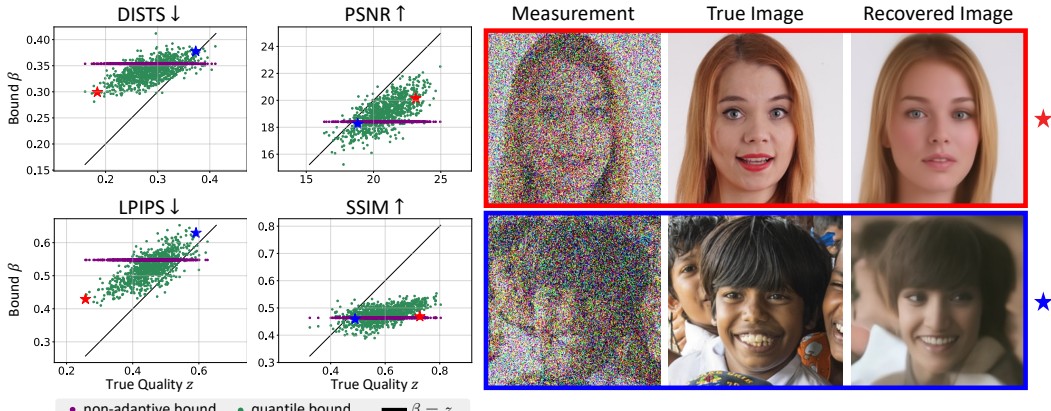

Figure 2: Scatter plots show the non-adaptive (purple) and quantile (green) bounds $\beta(\widehat{z}_k, \widehat{\lambda}(d_{\mathsf{cal}}[t]))$ versus the true FRIQ $z_k$ over FFHQ test samples $k$. The black line shows where $\beta = z$, and a fraction $\alpha = 0.05$ of samples are on the side of the line that violates the bound. The quantile bound tracks the true $z_k$ much better than the non-adaptive bound. The red and blue stars correspond to the images in the red and blue boxes: the red recovery represents better FRIQs and blue represents worse.

### 3.5 Constructing the image estimate $\widehat{x}$

As described above in Secs. 3.2–3.4, a posterior-sampling-based image recovery method allows one to construct adaptive bounds using image samples $\{\widetilde{x}_i^{(j)}\}_{j=1}^c$. But, as we now discuss, a posterior-sampling-based image recovery method also provides flexibility in how $\widehat{x}_i$ itself is constructed.

For example, when one is interested in constructing $\widehat{x}_i$ with high PSNR, or equivalently low MSE, it makes sense to set $\widehat{x}_i$ as the minimum MSE (MMSE) or conditional-mean estimate $\mathrm{E}\{X_i|Y_i=y_i\}$. This can be approximated by the empirical mean of $p$ posterior samples, i.e.,

$$\widehat{x}_i = \frac{1}{p} \sum_{j=c+1}^{c+p} \widetilde{x}_i^{(j)}, \tag{11}$$

with large $p$. The indices on $j$ in (11) are chosen to avoid the samples $\{\widetilde{x}_i^{(j)}\}_{j=1}^c$ used for the adaptive bounds. However, because the MMSE estimate can look unrealistically smooth, smaller values of $p$ are appropriate when constructing an $\widehat{x}_i$ with good SSIM, DISTS, or LPIPS performance. For example, (Bendel et al., 2023) found that, for multicoil brain MRI at acceleration $R = 8$ with a particular posterior sampler, the best choice of $p$ is 8 for SSIM and 2 for both DISTS and LPIPS. This is can explained by the perception-distortion tradeoff (Blau & Michaeli, 2018), which says that, as $p$ increases and the MSE distortion decreases, the perceptual quality must also decrease. In the end, each FRIQ metric prefers a particular tradeoff between perceptual quality and distortion.

## 4 Numerical experiments

We now consider two imaging inverse problems: image denoising and accelerated MRI. For each, we evaluate the proposed bounds using the PSNR, SSIM (Wang et al., 2004b), LPIPS (Zhang et al., 2018), and DISTS (Ding et al., 2020a) metrics.

### 4.1 Denoising

**Data:** For true images, we use a random subset of 4000 images from the Flickr Faces HQ (FFHQ) (Karras et al., 2019) validation dataset, to which we added white Gaussian noise of standard deviation $\sigma = 0.75$ to create the measurements $y_0$. The first 1000 images were used to train the predictor $f(\cdot; \theta)$ in (9) and the remaining 3000 were used for calibration and testing.

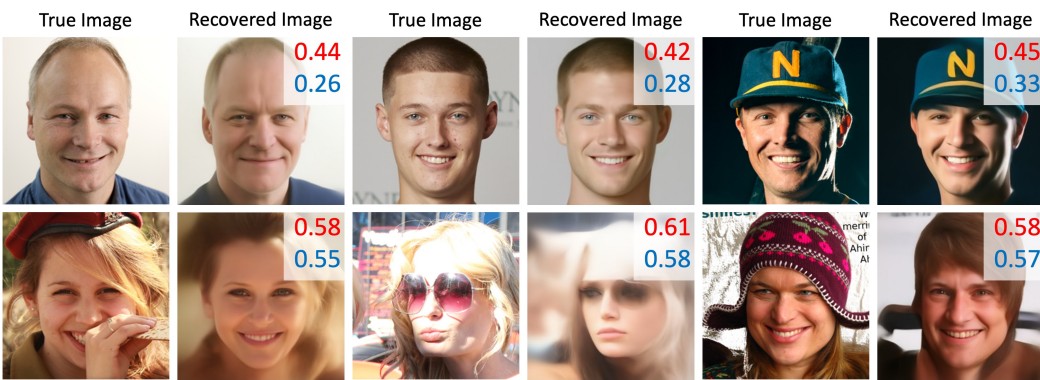

Figure 3: Examples from the FFHQ denoising experiment. Top row: true image and low-LPIPS recovery. Bottom row: true image and high-LPIPS recovery. True LPIPS reported in blue and quantile upper-bound in red. (Recall that LPIPS assigns lower values to better recoveries.)

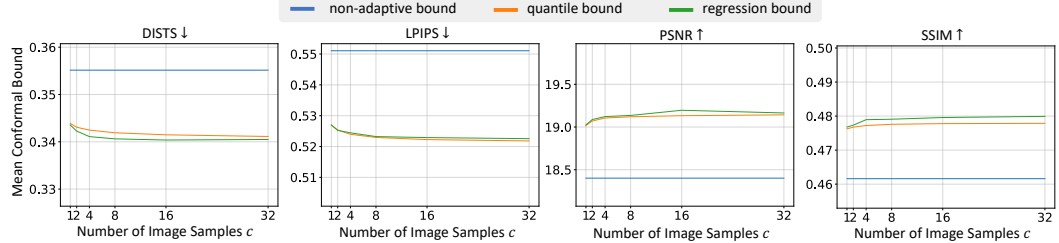

Figure 4: Mean conformal bound versus number of posterior samples $c$ for FFHQ denoising.

**Recovery:** To recover $\widehat{x}_0$ from $y_0$, a denoising task, we use the Denoising Diffusion Restoration Model (DDRM) (Kawar et al., 2022a). Following (Kawar et al., 2022a), we run DDRM with a Denoising Diffusion Probabilistic Model (DDPM) (Ho et al., 2020) pretrained on the CelebA-HQ dataset (Karras et al., 2018). To increase sampling diversity, we used $\eta = 1$ and $\eta_b = 0.5$ but set all other hyperparameters at their default values. For each measurement $y_i$, we use one DDRM sample (i.e., $p = 1$) for the image estimate $\widehat{x}_i$ and $c$ independent samples for $\{\widetilde{x}_i^{(j)}\}_{j=1}^c$.

**Conformal bounds:** We evaluate the proposed bounding methods from Secs. 3.1, 3.3, and 3.4, which we refer to as the **non-adaptive**, **quantile**, and **regression** bounds, respectively. For the regression bound, we use a quantile predictor $f(\cdot, \theta)$ that takes the form of a linear spline with two knots (see Appendix F for more details).

**Validation procedure:** Because the coverage guarantee (6) involves random calibration data and test data, we evaluate our methods using $T$ Monte-Carlo trials. For each trial $t \in \{1, \ldots, T\}$, we randomly select 70% of the 3000 non-training samples to create the calibration set $d_{\mathsf{cal}}[t]$ and we use the remaining 30% of the non-training samples for testing. In particular, we compute $\widehat{\lambda}$ using $d_{\mathsf{cal}}[t]$ and then, for each sample $k$ in the testing fold for trial $t$, we compute the bound $\beta(\widehat{z}_k, \widehat{\lambda}(d_{\mathsf{cal}}[t]))$. Finally, $\beta(\widehat{z}_k, \widehat{\lambda}(d_{\mathsf{cal}}[t]))$ is averaged over the test samples $k$ and the Monte Carlo trials $t$ to yield the "mean conformal bound" (MCB). Unless specified otherwise, we used error rate $\alpha = 0.05$, $T = 10\,000$, and $c = 32$ samples for the adaptive bounds. Appendix A demonstrates that all empirical coverages align with the theoretical guarantees in (6).

**Bound versus true metric:** Figure 2 shows scatter plots of the non-adaptive and quantile bounds $\beta(\widehat{z}_k, \widehat{\lambda}(d_{\mathsf{cal}}[t]))$ versus the true FRIQ $z_k$ for the test samples $k$ of a single Monte Carlo trial, along with the true image $x_k$ and recovery $\widehat{x}_k$ for two test samples. The sample highlighted in red has better subjective visual quality compared to the one in blue, and this is reflected in both the true FRIQ metrics $z_k$ and the corresponding quantile bounds, but not the non-adaptive bound. In Fig. 3, we show six additional samples from the FFHQ denoising experiment, three with low (true) LPIPS and

three with high (true) LPIPS, along with the respective true images. The quantile upper-bound on LPIPS is superimposed on each recovery. We see that the bounds are valid in the sense that they did not under-predict the true LPIPS, and adaptive in the sense that the bounding value is lower when the true LPIPS is lower.

**MCB versus bounding method and number of posterior samples $c$:** Figure 4 plots MCB versus the number of posterior samples $c$ used for the adaptive bounds. The figure shows that the non-adaptive bound is looser (i.e., smaller for the HP metrics PSNR and SSIM and larger for the LP metrics DISTS and LPIPS) than the two adaptive bounds. For both adaptive bounds, Fig. 4 shows only minor bound improvement with increasing $c$, suggesting that the adaptive bounds are robust to the choice of $c$, and that small values of $c$ could suffice if sample-generation was computationally expensive.

Interestingly, Fig. 4 shows relatively little improvement when going from the quantile bound to the regression bound. This may be due to our choice of a linear spline with two knots for $f(\cdot;\theta)$, but experiments with higher spline orders and/or more knots did not yield improved results, and neither did experiments with XGBoost (Chen & Guestrin, 2016) models for $f(\cdot;\theta)$. Additional experiments that hold the number of test samples at 900 and vary $n_{\mathsf{train}}$ and $n_{\mathsf{cal}}$ such that $n_{\mathsf{train}} + n_{\mathsf{cal}} = 3100$ (see Appendix B) also show little change in the performance of the quantile and regression bounds. Thus, for our experimental data, the effort to train the estimation function $f(\cdot;\theta)$ from (9) may not be justified, given the good performance of the simple empirical-quantile estimation function $f(\cdot)$ from (8). But the behavior may be different with other datasets.

**Computation time:** Computing a single DDRM sample takes approximately 2.73 seconds. Once the calibration constant $\widehat{\lambda}(d_{\mathsf{cal}})$ is known, computing $c = 32$ FRIQ samples $\{\widehat{z}_0^{(j)}\}_{j=1}^c$ and $\beta(\widehat{z}_0, \widehat{\lambda}(d_{\mathsf{cal}}))$ takes around 217ms, 320ms, 5ms, and 6ms for DISTS, LPIPS, PSNR, and SSIM, respectively. All times pertain to a single NVIDIA V100 with 32GB of memory.

## 4.2 ACCELERATED MRI

We now simulate our methods on accelerated multicoil MRI (Knoll et al., 2020; Hammernik et al., 2023). MRI is a medical imaging technique known for excellent soft tissue contrast without subjecting the patient to harmful ionizing radiation. MRI has slow scan times, though, which reduce patient throughput and comfort. In accelerated MRI, one collects only $1/R$ of the measurements specified by the Nyquist sampling theorem, thus speeding up the acquisition process by rate $R$. For $R > 1$, however, the inverse problem may become ill-posed, in which case one may be interested in bounding the FRIQ of the recovered image.

**Data:** We utilize the non-fat-suppressed subset of the multicoil fastMRI knee dataset (Zbontar et al., 2018), yielding 17286 training images and 2188 validation images. To simulate the imaging process, we retrospectively sub-sample in the spatial Fourier domain (the "k-space") using random Cartesian masks that give acceleration rates $R \in \{16, 8, 4, 2\}$. See App. E for additional details.

**Recovery:** For the quantile bound, we generate approximate posterior samples using the conditional normalizing flow (CNF) from (Wen et al., 2023a). We use $p$ samples to construct $\widehat{x}_i$ via (11) and $c$ additional samples $\{\widetilde{x}_i^{(j)}\}_{j=1}^c$ to construct the quantile bound. For the non-adaptive bound, we construct $\widehat{x}_i$ using the state-of-the-art E2E-VarNet (Sriram et al., 2020a), which is a deterministic reconstruction approach. Both methods are trained to work well with all four acceleration rates $R$. (See App. F for training details.) Similar to Sec. 4.1, we found that the regression bound did not provide significant gain over the quantile bound and so, to streamline our discussion, we consider only the quantile and non-adaptive bounds for MRI. As before, we evaluate performance over $T = 10\,000$ Monte Carlo trials with a random 70% calibration and 30% test split of the validation data. All experiments use an error-rate $\alpha = 0.05$. Methods are separately calibrated for each acceleration rate.

**Bound versus true-metric:** Figure 5 shows scatter plots of the true FRIQ $z_k$ versus the non-adaptive and quantile bounds $\beta(\widehat{z}_k, \widehat{\lambda}(d_{\mathsf{cal}}[t]))$ for the test samples $k$ in a single Monte-Carlo trial. The results are shown for $R = 8$ acceleration, $c = 32$ samples in the adaptive bounds, and the best performing $p$ for each metric (see App. C). Except for a few outliers, the quantile bound closely tracks the true FRIQ $z_k$, demonstrating good adaptivity, while the non-adaptive bounds remain constant with $z_k$.

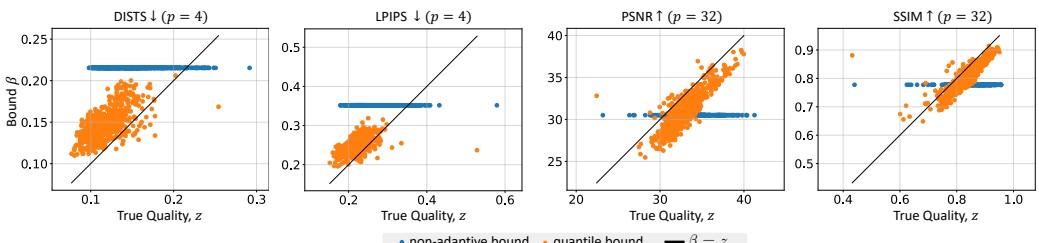

Figure 5: Scatter plots show the non-adaptive (blue) and quantile (orange) bounds $\beta(\widehat{z}_k, \widehat{\lambda}(d_{\mathsf{cal}}[t]))$ versus the true FRIQ $z_k$ over MRI test samples $k$ at acceleration $R = 8$. The black line shows where $\beta = z$. A fraction of $\alpha = 0.05$ samples are on the side of the line that violates the bound. Note that the quantile bound tracks the true $z_k$ much better than the non-adaptive bound.

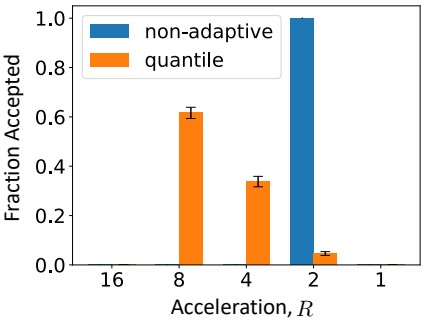

Figure 6: Fraction of accepted slices versus final acceleration rate for multi-round MRI. Error bars show standard deviation.

Table 1: Average results for a multi-round MRI simulation where measurement collection stop once bounds are below a user-set threshold $\tau$. Results shown for $T = 10\,000$ trials using the DISTS metric with $\alpha = 0.05$, $\tau = 0.15$, $p = 4$, and $c = 32$ ($\pm$ standard error).

| Method | Average Acceleration | Acceptance Empirical Coverage |
|---|---|---|
| Non-adaptive | $2.000 \pm 0.000$ | $0.9505 \pm 0.0001$ |
| Quantile | $5.422 \pm 0.001$ | $0.9461 \pm 0.0001$ |

**Multi-round Measurement:** To showcase the practical impact of our bounds, we adapt the multi-round measurement protocol from (Wen et al., 2024), where measurements are collected over multiple rounds until the uncertainty bound falls below a threshold. In our setting, measurements are first collected at acceleration $R = 16$, an image recovery is computed, and a conformal upper-bound on its DISTS is computed. If the bounding value is lower than a pre-determined threshold $\tau$, signifying that the recovery is (with probability $1 - \alpha$) of sufficient diagnostic quality (Kastryulin et al., 2023), then measurement collection stops. If not, additional measurements are collected and combined with the previous ones to yield an acceleration of $R = 8$, and the process repeats. We allow up to five measurement rounds, corresponding to final accelerations of $R \in \{16, 8, 4, 2, 1\}$.

Once again, we report average results across $T = 10\,000$ trials. Figure 6 plots the fraction of test image slices accepted by the multi-round protocol at each acceleration rate $R$. With the quantile bound, the measurements stop after two rounds (i.e., $R = 8$) in more than 60% of the cases, and after three rounds (i.e., $R = 4$) in more than 30% of the cases. With the non-adaptive bound, the measurements stop after four rounds (i.e., $R = 2$) in all cases. Table 1 shows that, with the quantile bound, the multi-round protocol attains an average acceleration of $R = 5.42$, which far surpasses the $R = 2$ acceleration achieved with the non-adaptive bound. Table 1 also shows that the empirical coverage of the multi-round accepted slices is very close to $1 - \alpha$, despite having only coverage guarantees (6) for a single-round measurement at each acceleration rate. Figure 7 shows examples of the image-error, the true DISTS, and its quantile upper-bound for each measurement round. With the threshold set at $\tau = 0.15$, the example on the top would collect two rounds of measurements (i.e., $R = 8$) while the example at the bottom would collect three rounds of measurements (i.e., $R = 4$), as demarcated by the red squares. See App. C for additional qualitative results.

**Computation time:** The E2E-VarNet takes approximately 104ms to generate a single posterior sample, while the CNF take about 2.41 seconds to generate 64 posterior samples (corresponding to

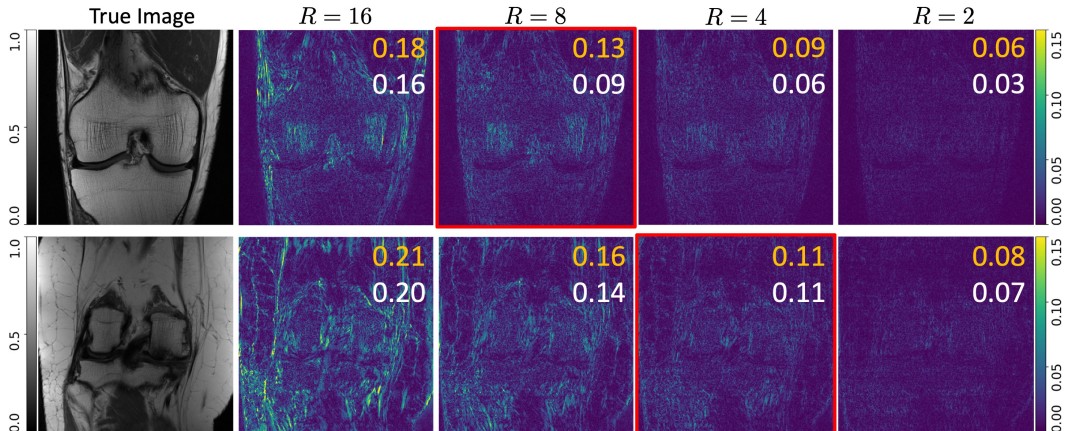

Figure 7: Examples of the multi-round MRI measurement procedure with DISTS at $\alpha = 0.05$, $\tau = 0.15$, $p = 4$, and $c = 32$. Error images at each acceleration $R$ are shown with the quantile bound (orange) and true metric (white). The red box indicates the measurement round at which the bound falls below the threshold $\tau$ and the measurement procedure concludes.

$p = 32$ and $c = 32$) on a single NVIDIA V100. The computation time of the metrics and bounds is on par with the times reported for the FFHQ experiments.

**Limitations:** We acknowledge multiple limitations in our proposed methodology. 1) Our methods require access to calibration data $\{(x_i, y_i)\}_{i=1}^n$ that is similar enough to the test data $(x_0, y_0)$ for the FRIQ pairs $\{(z_i, \hat{z}_i)\}_{i=0}^n$ to be modeled as statistically exchangeable. More work is required to make our methods robust to distribution shift (see App. D), although (Tibshirani et al., 2019; Barber et al., 2023; Cauchois et al., 2024) suggest some paths forward. 2) Our methods will be most impactful when there exists evidence that the FRIQ metric is well matched to the application (e.g., DISTS for MRI (Kastryulin et al., 2023)). For some applications, additional work is required to determine which metrics are more appropriate. 3) Our MRI application ideas are preliminary and not ready for practical use; rigorous clinical trials are needed to tune and validate the methodology on a much larger and diverse cohort of data. 4) The learned adaptive bound from Sec. 3.4 requires training a quantile regression model, and our FFHQ denoising experiment suggests that it may not be easy to significantly outperform the simpler adaptive bound from Sec. 3.3. 5) The posterior samplers that we considered in our numerical experiments target only aleatoric uncertainty, and sharper conformal bounds might be attained if epistemic uncertainty was also considered (e.g., (Ekmekci & Cetin, 2023)). 6) Because our methods are based on CP (or, equivalently, conformal risk control under the indicator loss (Angelopoulos et al., 2022a)), the marginal guarantee (6) holds with probability $1 - \alpha$ over random test data (e.g., $\hat{Z}_0, Z_0$) and calibration sets $D_{\text{cal}}$. A more fine-grained coverage could be achieved via the Risk-Controlling Prediction Sets (RCPS) framework from (Bates et al., 2021), which employs *two* user-selected error rates $\alpha, \delta \in (0, 1)$ to yield coverage guarantees like

$$\Pr\left[\Pr\left\{Z_0 \in \mathcal{C}_{\hat{\lambda}(D_{\text{cal}})}(\hat{Z}_0) \mid D_{\text{cal}}\right\} \geq 1 - \alpha\right] \geq 1 - \delta \tag{12}$$

in place of (6). In (12), $\alpha$ controls the $D_{\text{cal}}$-conditional error while $\delta$ controls the error over $D_{\text{cal}}$.

## 5 CONCLUSION

For imaging inverse problems, we used conformal prediction to construct bounds on the FRIQ of a recovered image relative to the unknown true image. When constructed using a calibration set that is statistically exchangeable with the test sample, our bounds are guaranteed to hold with high probability. Two of our methods leveraged approximate-posterior-sampling schemes to yield tighter conformal bounds that adapt to the measurements and reconstruction. Our approaches were demonstrated on image denoising and accelerated multicoil MRI, illustrating the broad applicability of our work.

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

Table 2: Mean empirical coverage for the non-adaptive method with $\alpha = 0.05$ and $T = 10\,000$ on the FFHQ denoising task ($\pm$ standard error)

| DISTS | LPIPS | PSNR | SSIM |
|---|---|---|---|
| $0.95000 \pm 0.00009$ | $0.95016 \pm 0.00009$ | $0.95004 \pm 0.00009$ | $0.95010 \pm 0.00009$ |

Table 3: Mean empirical coverage for the quantile method with $\alpha = 0.05$ and $T = 10\,000$ on the FFHQ denoising task ($\pm$ standard error)

| $c$ | DISTS | LPIPS | PSNR | SSIM |
|---|---|---|---|---|
| 1 | $0.95002 \pm 0.00009$ | $0.94997 \pm 0.00009$ | $0.95013 \pm 0.00009$ | $0.94989 \pm 0.00009$ |
| 2 | $0.95006 \pm 0.00009$ | $0.95003 \pm 0.00009$ | $0.95001 \pm 0.00009$ | $0.95022 \pm 0.00009$ |
| 4 | $0.94997 \pm 0.00009$ | $0.95008 \pm 0.00009$ | $0.94986 \pm 0.00009$ | $0.94999 \pm 0.00009$ |
| 8 | $0.95020 \pm 0.00009$ | $0.95015 \pm 0.00009$ | $0.95019 \pm 0.00009$ | $0.94991 \pm 0.00009$ |
| 16 | $0.94998 \pm 0.00009$ | $0.94999 \pm 0.00009$ | $0.95009 \pm 0.00009$ | $0.95008 \pm 0.00009$ |
| 32 | $0.95002 \pm 0.00009$ | $0.95013 \pm 0.00009$ | $0.95003 \pm 0.00009$ | $0.95006 \pm 0.00009$ |

Table 4: Mean empirical coverage for the regression method with $\alpha = 0.05$ and $T = 10\,000$ on the FFHQ denoising task ($\pm$ standard error)

| $c$ | DISTS | LPIPS | PSNR | SSIM |
|---|---|---|---|---|
| 1 | $0.94994 \pm 0.00009$ | $0.94970 \pm 0.00009$ | $0.95009 \pm 0.00009$ | $0.95014 \pm 0.00009$ |
| 2 | $0.95011 \pm 0.00009$ | $0.94953 \pm 0.00009$ | $0.94985 \pm 0.00009$ | $0.95004 \pm 0.00009$ |
| 4 | $0.94996 \pm 0.00009$ | $0.94946 \pm 0.00009$ | $0.95003 \pm 0.00009$ | $0.94995 \pm 0.00009$ |
| 8 | $0.95004 \pm 0.00009$ | $0.94964 \pm 0.00009$ | $0.94999 \pm 0.00009$ | $0.95017 \pm 0.00009$ |
| 16 | $0.94986 \pm 0.00009$ | $0.94964 \pm 0.00009$ | $0.95007 \pm 0.00009$ | $0.94987 \pm 0.00009$ |
| 32 | $0.95013 \pm 0.00009$ | $0.95026 \pm 0.00009$ | $0.95001 \pm 0.00009$ | $0.95006 \pm 0.00009$ |

## A  EMPIRICAL COVERAGE

In this section, we verify that the marginal coverage guarantee in (6) holds as expected. For each Monte Carlo trial $t$, we compute the empirical coverage for the non-adaptive method as

$$\text{EC}[t] \triangleq \frac{1}{|\mathcal{I}_{\text{test}}[t]|} \sum_{i \in \mathcal{I}_{\text{test}}[t]} \mathbb{1}_{z_i \in \mathcal{C}_{\widehat{\lambda}(D_{\text{cal}})}}, \tag{13}$$

where $\mathcal{I}_{\text{test}}[t]$ is the set of indices for the test samples of trial $t$. Computing the empirical coverage for the adaptive methods can be done in the same manner with the appropriate $\mathcal{C}_{\widehat{\lambda}(D_{\text{cal}})}(\cdot)$. In Tables 2, 3, 4, we report the average empirical coverage and standard error across $T = 10\,000$ trials for all three methods on the FFHQ experiments using $\alpha = 0.05$. For all methods, the average empirical coverage is very close to the theoretical coverage $1 - \alpha = 0.95$ regardless of the metric or value of $c$, demonstrating close adherence to the theory. There are very slight deviations as a result of finite trials, number of calibration samples, and number of testing samples.

In Table 5, we report the mean empirical coverage for the quantile method in the MRI experiments with $R = 8$, $\alpha = 0.05$, $c = 32$, and $p \in \{1, 2, 4, 8, 16, 32\}$ across $T = 10\,000$ trials. For any value of $p$, we see the empirical coverage is very close to the theoretical $1 - \alpha = 0.95$ coverage; thus, once again, our method shows close compliance to the theory.

## B  ADDITIONAL FFHQ DENOISING EXPERIMENTS

**Effect of training and calibration set size:**  For FFHQ denoising, we now investigate how the amount of training and calibration data affect the mean conformal bound. Following the same Monte Carlo procedure as Sec. 4.1, we fix the number of testing samples to 900 but change the proportion of $n_{\text{train}}$ versus $n_{\text{cal}}$ for the remaining 3100 samples. In Fig. 8, we show the mean conformal bounds

Table 5: Mean empirical coverage for the quantile method with $\alpha = 0.05$, $c = 32$, and $T = 10\,000$ on the $R = 8$ accelerated MRI task ($\pm$ standard error). All coverages are above the expected coverage of $1 - \alpha = 0.95$

| $p$ | DISTS | LPIPS | PSNR | SSIM |
|---|---|---|---|---|
| 1 | $0.9503 \pm 0.0001$ | $0.9503 \pm 0.0001$ | $0.9505 \pm 0.0001$ | $0.9504 \pm 0.0001$ |
| 2 | $0.9505 \pm 0.0001$ | $0.9503 \pm 0.0001$ | $0.9504 \pm 0.0001$ | $0.9505 \pm 0.0001$ |
| 4 | $0.9503 \pm 0.0001$ | $0.9503 \pm 0.0001$ | $0.9505 \pm 0.0001$ | $0.9504 \pm 0.0001$ |
| 8 | $0.9505 \pm 0.0001$ | $0.9504 \pm 0.0001$ | $0.9504 \pm 0.0001$ | $0.9505 \pm 0.0001$ |
| 16 | $0.9505 \pm 0.0001$ | $0.9502 \pm 0.0001$ | $0.9504 \pm 0.0001$ | $0.9504 \pm 0.0001$ |
| 32 | $0.9504 \pm 0.0001$ | $0.9506 \pm 0.0001$ | $0.9504 \pm 0.0001$ | $0.9505 \pm 0.0001$ |

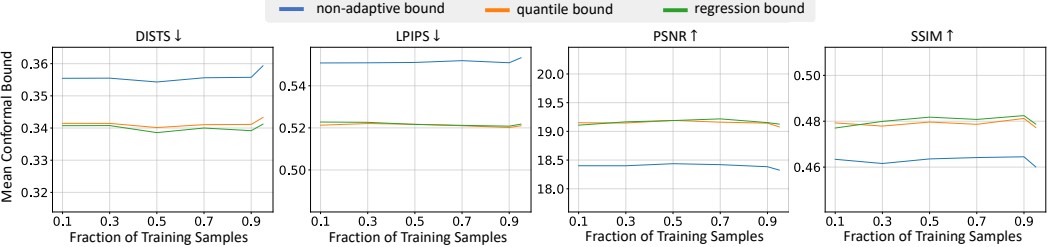

Figure 8: Mean conformal bound versus the proportion of training samples for FFHQ denoising with $n_{\mathsf{train}} + n_{\mathsf{cal}} = 3100$ samples.

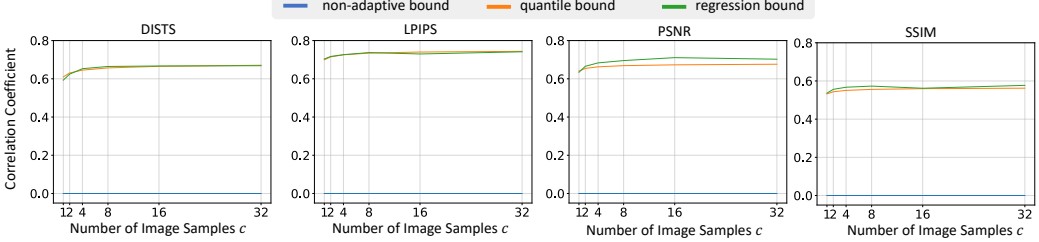

Figure 9: Mean Pearson correlation coefficient between each conformal bound and the true FRIQ versus the number of posterior samples $c$.

as the proportion of training samples varies, starting with $0.1$ and going up to $0.95$, for $T = 10\,000$, $c = 32$, and $\alpha = 0.05$. Both adaptive methods still provide noticeable gains over the non-adaptive bound. Even with additional training samples, however, the regression bounds show relatively little improvement over the quantile bounds. Based on (3), the conformal bounds should grow more conservative as the number of calibration points decreases for the non-adaptive and quantile bounds. However, this effect is not evident until very small calibration set sizes (e.g., when the fraction of calibration samples is $0.05$).

**Correlation between conformal bound and true FRIQ:** Figure 2 visually demonstrates that the quantile bound tracks the true FRIQ much better than the non-adaptive bound. To quantify this tracking behavior, we compute the Pearson correlation coefficient between each conformal bound $\beta(\widehat{z}_k, \widehat{\lambda}(d_{\mathsf{cal}}[t]))$ and the true FRIQ $z_k$ over the test samples $k$ for each Monte-Carlo trial $t$. In Fig. 9, we plot the mean (across $T = 10000$ trials) Pearson correlation coefficient versus $c$ for each bound. Since the non-adaptive bound is constant with $z_k$, its correlation equals 0. However, the two adaptive approaches demonstrate a correlation coefficient above $0.5$, and up to $0.7$, depending on the metric. These correlation coefficients quantify the adaptivity of our bounds and explain, in part, why the adaptive bounds led to better average acceleration rates than the non-adaptive bound in the multi-round measurement experiment of Sec. 4.2.

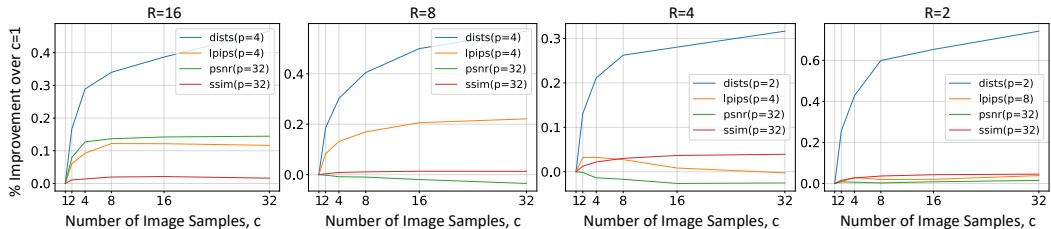

Figure 10: Percent improvement in MCB versus number of samples $c$ used in the quantile bound. For each metric, we use the value of $p$ that gave the best MCB at $c = 32$, as denoted in the legend.

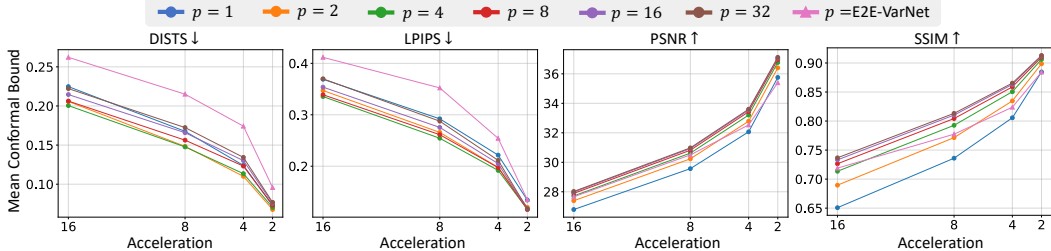

Figure 11: Mean conformal bound versus acceleration $R$ for accelerated MRI. The non-adaptive bound uses the E2E-VarNet for $\widehat{x}$ and the quantile bound averages $p$ CNF samples for $\widehat{x}$. Various $p$ shown.

## C  ADDITIONAL MRI EXPERIMENTS

**Effect of number of posterior samples $c$ in conformal bound:** For the case of FFHQ denoising, Sec. 4.1 demonstrated the number of posterior samples $c$ has a limited effect on the conformal bounds for the FFHQ experiments. We now investigate whether the same occurs with MRI. Figure 10 plots MCB versus $c$ using the value of $p$ (i.e., the number of posterior samples averaged to form $\widehat{x}$) that maximizes MCB when $c = 32$. From the figure, we see less than a $1\%$ improvement over $c$ for any metric, suggesting that the quantile method is indeed robust to the choice of $c$ for both experiments.

**Effect of acceleration rate $R$:** Figure 11 plots MCB versus acceleration rate $R$ with $c = 32$ and several values of $p$. In all cases, MCB improves as the acceleration $R$ decreases, as expected. However, as discussed in Sec. 3.5, each metric benefits from a different choice of $p$. DISTS and LPIPS prefer $p \in \{2, 4\}$ while PSNR and SSIM prefer $p = 32$. The figure also shows that, for each metric, the MCB for the E2E-VarNet-based method is worse than the MCB for the $p$-optimized CNF-based method, even though App. G shows that E2E-VarNet's average PSNR and SSIM scores are better. This further demonstrates the advantages of posterior-sampling-based image recovery.

**Multi-round measurement samples:** In Fig. 12, we show the zero-filled measurement, recovered image, and absolute-error map at each acceleration rate for $p = 4$. The conformal bound is imposed on the reconstructions for the case when $\alpha = 0.05$, $\tau = 0.15$, and $c = 32$. Following the multi-round measurement protocol described in Sec. 4.2, the reconstruction at $R = 8$ (marked in red) would be deemed sufficient ($\beta_i < \tau$), and the measurement collection would end.

## D  EMPIRICAL INVESTIGATION OF DISTRIBUTION SHIFT

As previously mentioned, a general limitation of CP methods like Angelopoulos et al. (2022a) is the requirement of exchangeability, which in our case applies to the pairs $\{(\widehat{Z}_i, Z_i)\}_{i=0}^n$. This requirement may be violated when there is a distributional shift between the test data $(x_0, y_0)$ and the calibration data $\{(x_i, y_i)\}_{i=1}^n$, which can then cause a distributional shift between the corresponding FRIQ quantities $(\widehat{z}_0, z_0)$ and $\{(\widehat{z}_i, z_i)\}_{i=1}^n$.

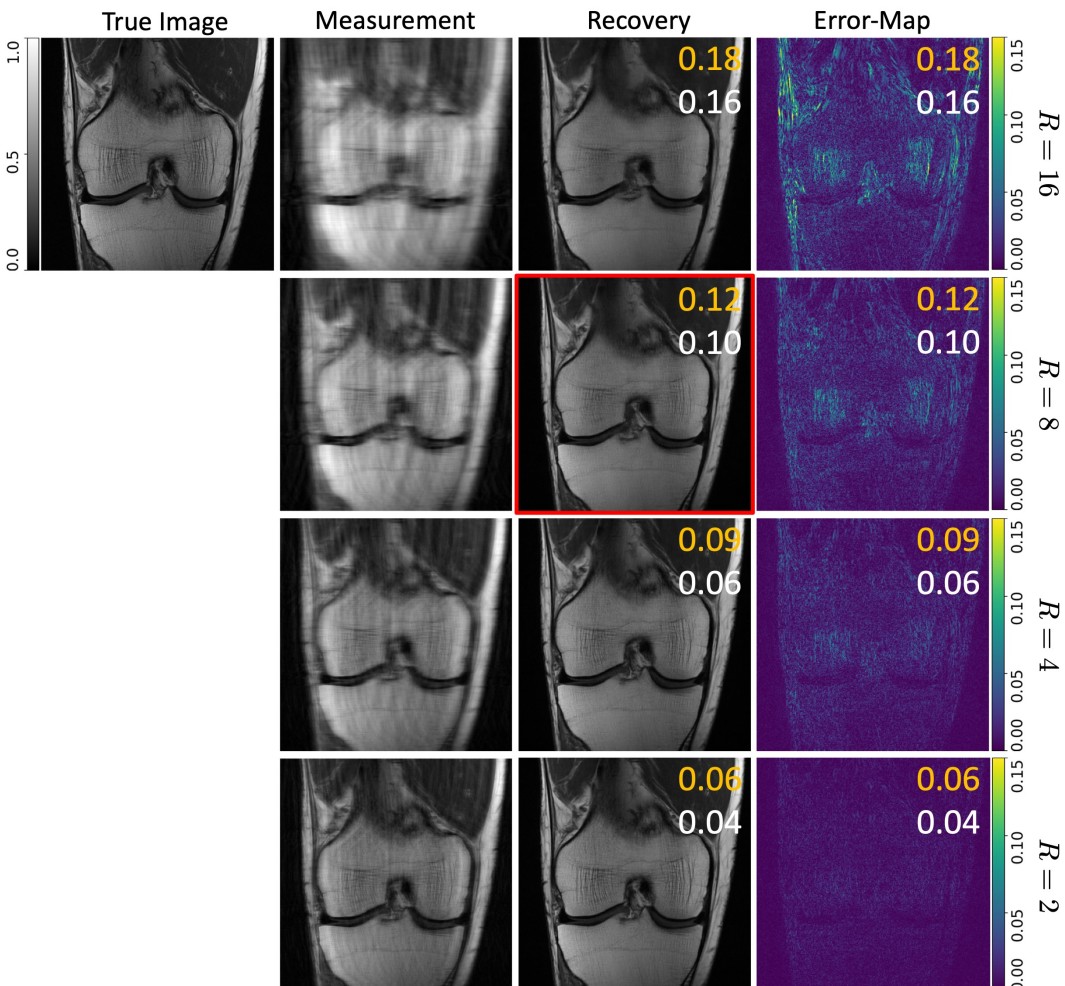

Figure 12: Qualitative example of the multi-round MRI experiment with DISTS at $\alpha = 0.05$, $\tau = 0.15$, $p = 4$, and $c = 32$. The measurement, recovery, and absolute error are shown for all accelerations. The quantile bound (orange) and true DISTS (white) are imposed on the reconstructions. The red box indicates the accepted reconstruction where the bound first falls below the threshold $\tau$.

In the case of MRI, such distributional shifts may arise for various reasons, some of which would be easy to prevent while others would be more difficult. For example, if the CP method was calibrated on knee images, one would not want to immediately test on brain images, but instead recalibrate a CP method on brain images. Likewise, if the CP method was calibrated with data from one manufacturer and/or strength of scanner, then it would be best to test on data from the same manufacturer and/or strength of scanner. Still, due to limited calibration data, situations may arise where a distribution shift is inevitable. Thus, we perform a study to analyze the sensitivity of our proposed method to distribution shifts.

For this study, we use the validation fold of the non-fat-suppressed multicoil fastMRI knee dataset Zbontar et al. (2018), which contains 100 3D volumes. A volume contains all the images collected for a single patient, with each image showing a different slice of the knee (from front to back). To induce a realistic yet controllable distribution shift, we choose calibration images from only the center slices of these volumes, and refer to the center slices as "location $l = 0$." We then create one test set with images from slice locations $l = 0$, another test set with images from slice location $l = 1$, and so on, until slice location $l = 10$ (which typically corresponds to an edge slice). Example images from various slice locations are shown in Fig. 13.

1026
1027
1028
1029
1030
1031
1032
1033
1034
1035
1036
1037
1038
1039
1040
1041
1042
1043
1044
1045
1046
1047
1048
1049
1050
1051
1052
1053
1054
1055
1056
1057
1058
1059
1060
1061
1062
1063
1064
1065
1066
1067
1068
1069
1070
1071
1072
1073
1074
1075
1076
1077
1078
1079

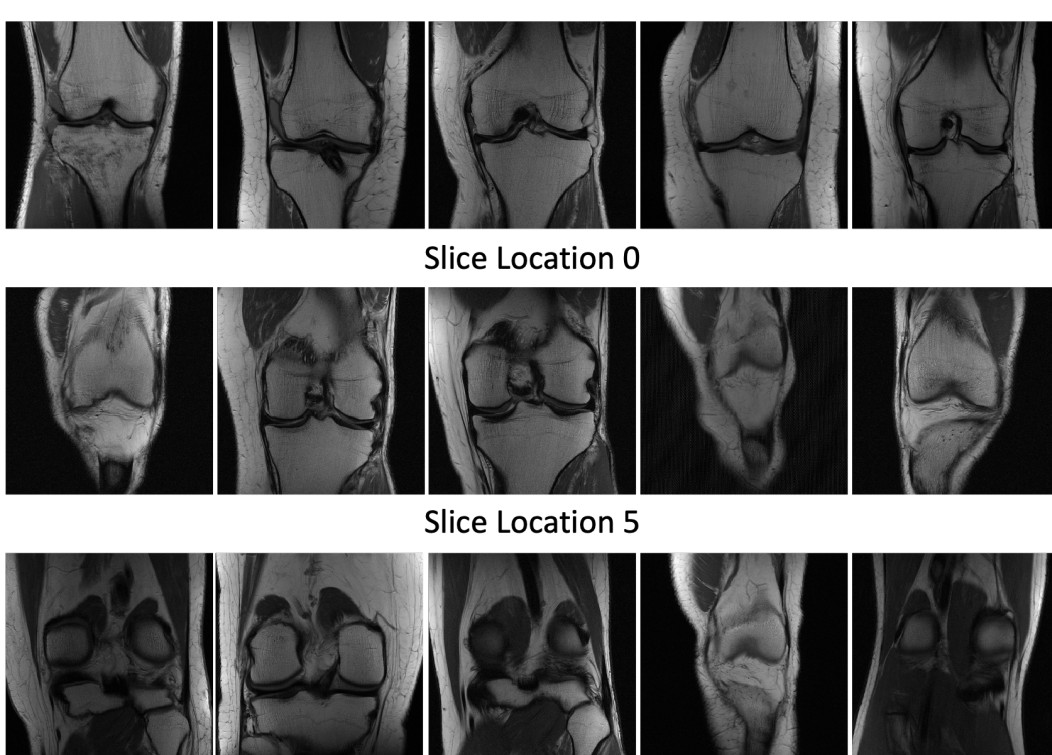

Slice Location 0

Slice Location 5

Slice Location 10

Figure 13: Qualitative examples of images from different slice locations. Slice location 0 indicates the center slice of a volume while larger slice locations are further towards the edges of a volume.

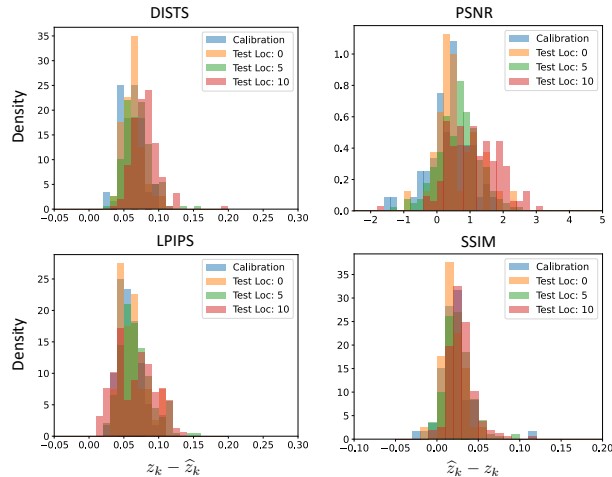

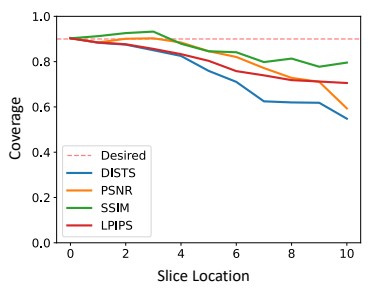

Figure 15: The average empirical coverage across $T = 10000$ trials for test sets at different slice locations. All trials are calibrated with images from slice location 0 with $\alpha = 0.1$, $R = 8$, $p = 1$, and $c = 32$.

Figure 14: Histograms of the difference between the true FRIQ $z_k$ and the FRIQ estimate $\widehat{z}_k$ for test samples $k$ in the test fold of a single trial. Histograms are shown for test slice locations $l = 0, 5, 10$. Note the increasing shift in distribution from the calibration set as $l$ increases.

We first evaluate the coverage of the quantile bound using $T = 10\,000$ Monte Carlo trials, error-rate $\alpha = 0.1$, acceleration $R = 8$, an average of $p = 1$ samples for $\widehat{x}_i$, and $c = 32$ posterior samples for $u_i$. For each trial $t \in \{1, \ldots, T\}$, we construct the calibration set by randomly sampling 60 of the 100 center slices. For the same $t$, we form the test data at location $l = 0$ using the remaining 40 slices, and we form the test data at locations $l > 0$ by randomly sampling 40 of the 200 available slices. Figure 15 plots the mean empirical coverage over the $T$ trials as a function of test slice location $l$. As expected, the desired $1 - \alpha$ coverage is met when $l = 0$, but the coverage tends to decrease as the slice location $l$ increases. Surprisingly, for the PSNR and SSIM metrics, the coverage remains close to or above $1 - \alpha$ up until $l = 4$, suggesting our method is robust to small distributional shifts with these metrics. For DISTS and LPIPS, the drop in coverage is less than 0.1 until after $l = 4$.

To visualize the distribution shift versus test location $l$, we consider the difference between the true FRIQ $z_k$ and the FRIQ estimate $\widehat{z}_k$ for each test sample $k$ in a single trial $t$. This difference is $z_k - \widehat{z}_k$ for LP metrics and $\widehat{z}_k - z_k$ for HP metrics. Figure 14 shows the histogram of this difference for test locations $l \in \{0, 5, 10\}$. As expected, these histograms deviate more as the test location $l$ increases, although the amount of deviation depends on the FRIQ metric. For PSNR, we see the histogram shifting to the right and widening, while for LPIPS, the histogram becomes bimodal at test location $l = 10$.

Figure 15 suggests that one could select a more conservative $\alpha$ to ensure sufficiently high coverage under small distributional shifts, but at the cost of more conservative bounds. In fact, this is largely the mechanism behind distributionally robust CP extensions like Cauchois et al. (2024). We leave such generalizations to future work.

## E   MRI SUBSAMPLING MASK DETAILS

For the MRI experiments, we simulate the collection of measurements at four acceleration rates $R = \{16, 8, 4, 2\}$. These measurements are collected in the 2D spatial frequency domain known as k-space, and the pattern with which these samples are collected is called a sampling mask.

For this study, we use a Cartesian sampling procedure where full lines of the 2D k-space are collected progressively. Starting with $R = 16$, we utilize a Golden Ratio Offset (GRO) (Joshi et al., 2022) sampling mask with GRO-specific parameters $s = 15$ and $\alpha = 8$. This gives a fully-sampled region of 9 lines in the center of k-space known as the autocalibration signal (ACS) region. To simulate the iterative collection of measurements, we build upon this mask for $R = 8$. We first collect central

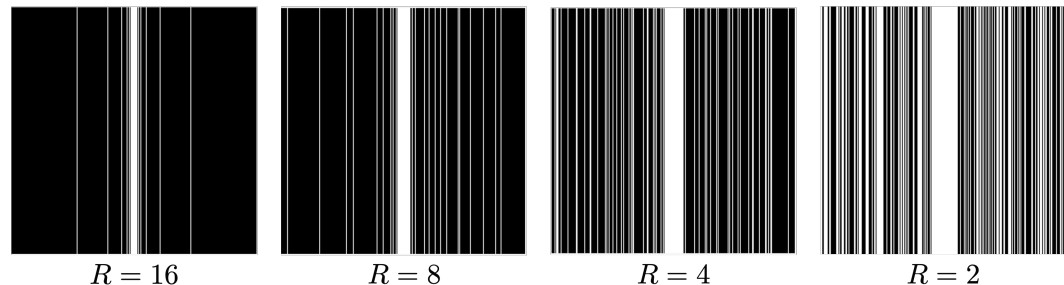

$$R = 16 \qquad R = 8 \qquad R = 4 \qquad R = 2$$

Figure 16: MRI sampling masks in k-space for each acceleration rate $R$. White pixels indicate the measurement was collected for that location in k-space. The masks are designed in a nested fashion where each mask contains all the measurements of higher $R$.

lines to obtain an ACS region of 16 lines before sampling additional k-space lines with a sampling probability inversely proportional to the distance from the center. Additional lines are collected until the desired acceleration $R = 8$ is met. This procedure is repeated for $R = 4$ and $R = 2$ to acquire masks with ACS widths of 24 and 32, respectively. Fig. 16 illustrates examples of the resulting masks.

## F TRAINING/MODEL DETAILS

For the regression bound in Sec. 3.4, where $u_i \in \mathbb{R}^c$, we use a quantile predictor of the form

$$f(u_i; \theta) = \psi(u_i)^\top w + b \ \text{ with } \ \theta = [w, b]^\top,, \tag{14}$$

where $\psi(\cdot)$ is a linear spline with two knots, $t_1$ and $t_2$, implemented via the truncated power basis

$$\psi(u_i) = [u_i; (u_i - t_1 \underline{1})_+; (u_i - t_2 \underline{1})_+] \in \mathbb{R}^{3c}, \tag{15}$$

with $\underline{1}$ the $c$-dimensional vector of ones and $(x)_+ \triangleq \max(x, 0)$. The two knots were placed at the $\frac{1}{3}$ and $\frac{2}{3}$ empirical quantiles of the mean training feature $\{\frac{1}{c}\sum_{j=1}^c u_i^{(j)}\}_{i=n+1}^{n+n_{\text{train}}}$, respectively. Essentially, for each feature in $u_i$, (14) implements a piece-wise-linear regression function with three distinct pieces. To promote consistency in $u_i = [u_{i1}, u_{i2}, \ldots, u_{ic}]^\top$ across different $i$, the spline function $\phi(\cdot)$ first sorts the values $\{u_{ij}\}_{j=1}^c$ within each $u_i$. For $\rho(\theta)$ in (10), we use ridge regularization on the weights $w$. The resulting (10) is a quadratic program, which can be optimized using any convex solver. To tune the regularization weight $\gamma$, we use $K$-fold cross validation with $K = 5$ folds and select the weight that provides the lowest mean pinball loss across the 5 folds.

For DDRM, we use the author's implementation (Kawar et al., 2022b), which is publicly available under an MIT license.

Both fastMRI reconstruction models were trained once with all four acceleration rates. For each sample in an epoch, one of the four sampling masks is randomly drawn, allowing the model to see each sample at a different acceleration throughout the training.

With the E2E-VarNet, we use the author's codebase (Sriram et al., 2020b), which is released under an MIT license. For training, we utilize the default hyperparameters provided by the authors for the model on the fastMRI knee leaderboard. The model was trained for 50 epochs with a batch size of 16 and learning rate of 0.0001 using SSIM (Wang et al., 2004a) as the loss function. This takes around 38 hours on a single NVIDIA V100 with 32GB of memory.

To train the CNF, we start with the author's implementation (Wen et al., 2023b) that is available under an MIT license. We modify the architecture slightly in order to better handle multiple accelerations. First, we include an invertible attention module, iMAP (Sukthanker et al., 2022), to the end of the base flow step. Then, we increase the number of initial channels in the conditioning network to 256. Using 2 layers and 10 flows steps in each layer, we train the CNF to minimize the negative log-likelihood objective. The model is trained for 150 epochs with batch size 8 and learning rate 0.0001. On a single NVIDIA V100, this takes around 335 hours.

Table 6: Average reconstruction performance on the fastMRI (Zbontar et al., 2018) knee validation set for $R = 16$ ($\pm$ standard error)

| Network | DISTS ↓ | LPIPS ↓ | PSNR ↑ | SSIM ↑ |
|---|---|---|---|---|
| E2E-VarNet | $0.209 \pm 0.001$ | $0.354 \pm 0.001$ | $\mathbf{30.301 \pm 0.043}$ | $\mathbf{0.807 \pm 0.001}$ |
| CNF ($p = 1$) | $0.183 \pm 0.001$ | $0.312 \pm 0.001$ | $28.244 \pm 0.039$ | $0.688 \pm 0.002$ |
| CNF ($p = 2$) | $0.167 \pm 0.001$ | $0.292 \pm 0.001$ | $29.091 \pm 0.039$ | $0.730 \pm 0.001$ |
| CNF ($p = 4$) | $\mathbf{0.165 \pm 0.001}$ | $\mathbf{0.287 \pm 0.001}$ | $29.588 \pm 0.039$ | $0.755 \pm 0.001$ |
| CNF ($p = 8$) | $0.173 \pm 0.001$ | $0.296 \pm 0.001$ | $29.862 \pm 0.039$ | $0.770 \pm 0.001$ |
| CNF ($p = 16$) | $0.184 \pm 0.001$ | $0.314 \pm 0.001$ | $30.006 \pm 0.039$ | $0.777 \pm 0.001$ |
| CNF ($p = 32$) | $0.193 \pm 0.001$ | $0.333 \pm 0.001$ | $30.080 \pm 0.039$ | $0.781 \pm 0.001$ |

Table 7: Average reconstruction performance on the fastMRI (Zbontar et al., 2018) knee validation set for $R = 8$ ($\pm$ standard error)

| Network | DISTS ↓ | LPIPS ↓ | PSNR ↑ | SSIM ↑ |
|---|---|---|---|---|
| E2E-VarNet | $0.151 \pm 0.001$ | $0.262 \pm 0.001$ | $\mathbf{33.459 \pm 0.047}$ | $\mathbf{0.864 \pm 0.001}$ |
| CNF ($p = 1$) | $0.136 \pm 0.000$ | $0.248 \pm 0.001$ | $30.796 \pm 0.044$ | $0.761 \pm 0.002$ |
| CNF ($p = 2$) | $\mathbf{0.118 \pm 0.000}$ | $0.225 \pm 0.001$ | $31.754 \pm 0.044$ | $0.799 \pm 0.001$ |
| CNF ($p = 4$) | $0.119 \pm 0.000$ | $\mathbf{0.219 \pm 0.001}$ | $32.329 \pm 0.043$ | $0.821 \pm 0.001$ |
| CNF ($p = 8$) | $0.128 \pm 0.001$ | $0.228 \pm 0.001$ | $32.650 \pm 0.043$ | $0.834 \pm 0.001$ |
| CNF ($p = 16$) | $0.138 \pm 0.001$ | $0.243 \pm 0.001$ | $32.819 \pm 0.043$ | $0.840 \pm 0.001$ |
| CNF ($p = 32$) | $0.145 \pm 0.001$ | $0.255 \pm 0.001$ | $32.907 \pm 0.043$ | $0.843 \pm 0.001$ |

To compute the quadratic program for Sec. 4.1, we use the qpsolver (Caron et al., 2024) package under a LGPL 3.0 license along with the CVXOPT (Andersen et al., 2023) package under a GNU General Public License.

We use the TorchMetrics (Borovec et al., 2022) package under the Apache 2.0 license to compute PSNR, SSIM, and LPIPS. We use the author's code at (Ding et al., 2020b) for DISTS under a MIT license. For multicoil MRI, we first compute the magnitude images using the "root-sum-of-squares" (RSS) (Roemer et al., 1990) before computing any metric. Since DISTS and LPIPS require a 3-channel image, we repeat the magnitude image for all three channels and normalize the values to be between 0 and 1 before computing either metric.

All models use the PyTorch (Paszke et al., 2019) framework with a custom license allowing open use. The E2E-VarNet and CNF are implemented using PyTorch Lightning (Falcon et al., 2019) under an Apache 2.0 license.

# G   AVERAGE FASTMRI RECONSTRUCTION PERFORMANCE

To get a sense of the average reconstruction performance for the accelerated MRI task, we report the average metrics for both the E2E-VarNet and CNF on the non-fat-suppressed subset of the fastMRI knee validation set. Results for acceleration rates $R = 16, 8, 4,$ and $2$ are shown in Tables 6, 7, 8, 9, respectively. The E2E-VarNet outperforms the CNF in PSNR and SSIM across all accelerations. The CNF, on the other hand, provides lower DISTS and LPIPS values in all cases other than for LPIPS at acceleration $R = 2$.

# H   DATASETS

The Flickr-Faces-HQ (FFHQ) (Karras et al., 2019) is publicly available under the Creative Commons BY-NC-SA 4.0 license. The fastMRI (Zbontar et al., 2018) datasets is available under a royalty-free license for internal research and educational purposes by the NYU fastMRI initiative. The providers have deidentified and manually inspected images and metadata for protected health information (PHI) as part of an IRB-approved study.

Table 8: Average reconstruction performance on the fastMRI (Zbontar et al., 2018) knee validation set for $R = 4$ ($\pm$ standard error)

| Network | DISTS ↓ | LPIPS ↓ | PSNR ↑ | SSIM ↑ |
|---|---|---|---|---|
| E2E-VarNet | $0.110 \pm 0.001$ | $0.181 \pm 0.001$ | $\mathbf{36.030 \pm 0.053}$ | $\mathbf{0.905 \pm 0.001}$ |
| CNF ($p = 1$) | $0.100 \pm 0.000$ | $0.191 \pm 0.001$ | $33.090 \pm 0.048$ | $0.826 \pm 0.001$ |
| CNF ($p = 2$) | $\mathbf{0.087 \pm 0.000}$ | $0.170 \pm 0.001$ | $34.073 \pm 0.048$ | $0.856 \pm 0.001$ |
| CNF ($p = 4$) | $0.090 \pm 0.000$ | $\mathbf{0.166 \pm 0.001}$ | $34.666 \pm 0.048$ | $0.873 \pm 0.001$ |
| CNF ($p = 8$) | $0.099 \pm 0.000$ | $0.171 \pm 0.001$ | $34.998 \pm 0.047$ | $0.882 \pm 0.001$ |
| CNF ($p = 16$) | $0.106 \pm 0.001$ | $0.178 \pm 0.001$ | $35.174 \pm 0.047$ | $0.887 \pm 0.001$ |
| CNF ($p = 32$) | $0.110 \pm 0.001$ | $0.184 \pm 0.001$ | $35.265 \pm 0.047$ | $0.889 \pm 0.001$ |

Table 9: Average reconstruction performance on the fastMRI (Zbontar et al., 2018) knee validation set for $R = 2$ ($\pm$ standard error)

| Network | DISTS ↓ | LPIPS ↓ | PSNR ↑ | SSIM ↑ |
|---|---|---|---|---|
| E2E-VarNet | $0.059 \pm 0.000$ | $\mathbf{0.094 \pm 0.001}$ | $\mathbf{39.692 \pm 0.060}$ | $\mathbf{0.947 \pm 0.001}$ |
| CNF ($p = 1$) | $0.059 \pm 0.000$ | $0.118 \pm 0.000$ | $36.810 \pm 0.054$ | $0.907 \pm 0.001$ |
| CNF ($p = 2$) | $\mathbf{0.054 \pm 0.000}$ | $0.105 \pm 0.000$ | $37.667 \pm 0.054$ | $0.923 \pm 0.001$ |
| CNF ($p = 4$) | $0.055 \pm 0.000$ | $0.100 \pm 0.000$ | $38.171 \pm 0.054$ | $0.931 \pm 0.001$ |
| CNF ($p = 8$) | $0.058 \pm 0.000$ | $0.099 \pm 0.000$ | $38.448 \pm 0.054$ | $0.935 \pm 0.001$ |
| CNF ($p = 16$) | $0.060 \pm 0.000$ | $0.099 \pm 0.000$ | $38.593 \pm 0.054$ | $0.937 \pm 0.001$ |
| CNF ($p = 32$) | $0.061 \pm 0.000$ | $0.099 \pm 0.000$ | $38.668 \pm 0.054$ | $0.939 \pm 0.001$ |

