# OpenReview forum: "Conformal Bounds on Full-Reference Image Quality for Imaging Inverse Problems"
_ICLR.cc/2025/Conference — Submitted to ICLR 2025_

### Official Review · Reviewer_P5sN · 2024-10-21

**Soundness:** 3
**Presentation:** 2
**Contribution:** 1
**Rating:** 3
**Confidence:** 4

**Summary:**

The paper presents a conformal prediction framework for constructing bounds on the recovery error with respect to a full-reference quality metric (FRIQ) in imaging inverse problems. The paper leverages split conformal prediction to build bounds that are guaranteed to hold marginally over exchangeable samples of the joint distribution of (measurement, reconstruction).
A posterior sampler is used to generate adaptive bounds (ie which change across measurements in the dataset),
The paper proposes to use a predictor (based on simple regression models, such as splines) to improve the quantile estimates, in order to reduce the number of posterior samples needed to obtain an accurate quantile.

**Strengths:**

- applies split conformal prediction to the linear inverse problems, with a generalization to FRIQ metrics.

**Weaknesses:**

Main weaknesses:
- the contribution of the paper is not novel enough in my opinion: the theory of split conformal prediction is defined with respect to any confirmity score, so I believe the use of general FRIQS as confirmity scores is not a very novel extension.
- the proposed quantile regression method seems to be the main methodological contribution of the paper, which is based on a relatively classical 1-dimensional interpolation method, and doesn't offer much improvement over a standard empirical quantile estimate.
- the image denoising results are not convincing: using a posterior sampler increases the computational fold 32 x diffusion_steps -fold with respect to an end-to-end reconstruction method, only to obtain a bound on the PSNR which is less than 1dB better than the non-adaptive bound (see figure 4).

A smaller weakness:
- the mathematical notation is often too heavy, rendering simple concepts hard to understand. Equation 13 is unclear: the dependence of $\bar{z}_i$ on $\theta$ is missing, $\lambda$ is the same symbol as used in equations 9 and 10 for calibrating the intervals, and here has another meaning.

**Questions:**

- why is the posterior mean in equation 17 computed using different samples than those used for approximating quantiles?
- why does the paper restrict the method to approximate posterior samplers? one could use any uncertainty quantification method to compute adaptive quantiles.

---

> ### Author Response · Authors · 2024-11-22
>
> We thank the reviewer for their time and feedback.
> Based on your feedback, we made several key modifications and submitted a revised version of our paper.
> The key changes include: 1) simplified notation in Secs. 2 and 3, and 2) the addition of a distribution-shift sensitivity study in App. D.
> All revisions are in colored text.
> Below we address your questions and concerns.
>
> ---
>
> **Weaknesses**
>
> 1) The contribution is not novel enough because split conformal can be applied to any non-conformity score.
>
> - First, we believe that the problem of estimating FRIQ metrics without knowing the ground-truth image is novel (Reviewers 8V4H,1Tr7,K6sy), significant (Reviewer 8V4H), useful in real life (Reviewer 1Tr7), and interesting to the computational imaging community (Reviewer K6sy). Second, there are many ways that one could go about estimating FRIQ metrics, and we believe that the choice to use split conformal prediction is itself novel. Third, although split conformal prediction can indeed be applied to any non-conformity score, there is considerable flexibility in *how* it is applied, and we believe that there is novelty in the particular way that we employ posterior image sampling to build the prediction intervals. Our manuscript has been revised and restructured to better showcase our contributions.
>
> 2) The proposed quantile regression method seems to be the main methodological contribution of the paper, and it doesn't offer much improvement over a standard empirical quantile estimate.
>
> - Response: Split-conformal prediction is a turn-the-crank procedure after the prediction interval $\\mathcal{C}\_\\lambda(\\cdot)$ has been designed. Our main contributions are the two methods we propose to construct $\\mathcal{C}\_\\lambda(\\cdot)$ from the measurements $y_0$; Sec. 3.2 presents our design intuitions while Secs. 3.3 and 3.4 provide the details of our designs. We agree that the learned regression bound is not much better than the empirical quantile bound, but we don't see this as a weakness. Rather, we believe that it attests to the accuracy of the design intuitions provided in Sec. 3.2, which establish that the ideal conformal bound can be computed using an empirical quantile of an infinite number of perfect-posterior samples. Our numerical experiments suggest, perhaps unsurprisingly, that relaxing the ideal scheme to use a finite number of approximate posterior samples is nearly as good as training a bound-predictor from scratch.
>
> 3) The image denoising results are not convincing. The adaptive bounds increase the compute significantly with only a 1dB gain in PSNR over the non-adaptive approach in Fig. 4.
>
> - Response: Fig. 4 reports only the *average* conformal bound across test samples. The adaptive bounds (and the true FRIQ) can be far above or below the average, depending on the test sample. The advantage of adaptivity can be better seen in Figs. 2 and 5, where the adaptive bound is looks much better correlated with the true PSNR than the non-adaptive bound, as well as in Table 1, where the adaptive bound results in an average accepted acceleration of 5.42 versus 2. In the revision, the newly added Fig. 9 shows the correlation coefficient between the conformal bound and the true FRIQ versus $c$. If inference speed was a concern, the diffusion sampler can be substituted with a faster posterior sampling approach, as we did with a normalizing flow in the MRI experiments.
>
> 4) The mathematical notation is often too heavy.  Also, Original Eq. 13 (Revised Eq. 10) is unclear due to a missing $\theta$ dependence and double usage of $\lambda$.
>
> - Response: Thank you for your feedback.  We've streamlined the notation in the revision, including adding a $\theta$ dependence to $\widehat{z}_i$ in that equation and avoiding a double meaning for $\lambda$.

---

> ### Author Response · Authors · 2024-11-22
>
> **Questions**
>
> 1) "Why is the posterior mean in Original eq. 17 (Revised eq. 11) computed using different samples than those used for approximating quantiles?"
>
> - Response: Consider the extreme case of $p=1$ in (11). If we used $\\widetilde{x}\_i^{(j)}$ to compute $\\widehat{x}\_i$, then $\\widetilde{z}\_i^{(j)}=m(\\widehat{x}\_i,\\widetilde{x}\_i^{(j)})=m(\\widetilde{x}^{(j)}\_i,\\widetilde{x}\_i^{(j)})$, which falsely indicates a perfect FRIQ. We believe that the design insights presented in Sec. 3.2 further explain why this would be a bad choice. Recall that, for an arbitrary fixed image estimate $\\widehat{x}\_i$, we use the FRIQ samples $\\{m(\\widehat{x}\_i,\\widetilde{x}\_i^{(j)})\\}_{j=1}^c$ as an empirical distribution that approximates the true distribution of $Z_0=m(\\widehat{x}\_i,X\_i)$ given $Y_0=y_0$.  Because $X_0$ is conditionally independent of $\\widehat{x}\_i$ given $Y_0=y_0$, we want the same to be true of $\\{\\widetilde{X}\_i^{(j)}\\}\_{j=1}^c$.
>
> 2) "Why does the paper restrict the method to approximate posterior samplers? One could use any uncertainty quantification method to compute adaptive quantiles."
>
> - Response: We use posterior image samplers because they are readily available.  The research community has invested a huge amount of effort into the design/implementation of those methods, and we capitalize on those efforts. That said, a non-sampling based approach could indeed be used to produce an estimate of the $\alpha$th quantile of the unknown true FRIQ $Z_0|Y_0=y_0$. For example, one could train the parameters $\varphi$ of a neural network $g(\widehat{x}_0;\varphi)$. But it's unclear what architecture to use, since nobody has ever designed such a network. Also, since $g(\widehat{x}_0;\varphi)$ takes in a high-dimensional image $\widehat{x}_0$, it would involve vastly more learnable parameters than our regression network $f(u_i;\theta)$, where $u_i$ has dimension 32 or less.

---

> > ### Comment · Reviewer_P5sN · 2024-11-27
> >
> > Mann thanks for your responses to my questions.
> >
> > After carefully considering your response, I prefer to keep my score, since my main concern of novelty remains.
> >
> > - I still believe that setting FRIQ = confirmity score, and applying split conformal prediction tools is not novel enough for an ICLR publication, especially in view of other papers I've reviewed for this conference.
> > - Using posterior samples to compute the bounds is not novel in my opinion, it is a fairly standard choice given the large number of papers in this area. The idea of using approximate posterior samples to construct adaptive bounds appears from the first works of Vovk on the topic.
> > - The utility of the proposed quantile regression method is not convincing to me, and the answer has not changed my mind unfortunately.
> > - The answer seems to suggest that the empirical posterior samplers can be close to the true posterior samples. I don't believe this is true, and is not demonstrated in any of the experiments of the paper (it is in fact not possible to demonstrate in high dimensional regression unless data is generated synthetically).

---

> > > ### Author Response · Authors · 2024-11-27
> > >
> > > We thank you for reviewing our response and continuing the discussion.
> > >
> > > First, we appreciated your feedback on how to improve our paper, and we would like to know if we have addressed your concerns with the notation and presentation.
> > > We put significant effort into improving the clarity of our method and making the notation easier to follow.
> > > If so, we hope you will consider raising your presentation score.
> > >
> > > Second, in regards to the concerns of novelty:
> > >
> > > 1) Reviewer: Setting FRIQ = conformity score and using conformal prediction is not novel enough.
> > >
> > > - Response: We encourage the reviewer to step back from the scope of conformal prediction and to view the problem from the perspective of computational imaging. We present the *first ever* approach to estimate FRIQ for computational imaging problems when the true image $x_0$ is unknown. Furthermore, our estimation approach comes with rigorous probabilistic guarantees. This is a significant contribution to the computational imaging community, and it's especially important in safety-critical applications. Furthermore, there is a real practical impact, as evidenced by the MRI multi-round measurement protocol in Sec. 4.2. It's true that we did not prove any new theorems for conformal prediction theory (and we don't claim to), but we don't believe that is a requirement for ICLR. We highlight that complexity does not always equate to novelty and often working from first principles with a new perspective can be effective.
> > >
> > > 2) Reviewer: Using posterior samples to compute bounds is not novel and the idea of approximate posterior samples to construct adaptive bounds appears from the first works of Vovk.
> > >
> > > - Response: While there have been many posterior sampling techniques proposed for computational imaging, there has been a gap in how to best utilize these posterior samples. We'd like to further emphasize that our main technical contributions are the design intuitions provided in Section 3.2, which describe exactly how posterior samples can be used to bound FRIQ. For example, one could easily make the mistake of computing the posterior mean in Eq. (11) using the same samples as those used for approximating quantiles (as the reviewer suggested in the initial review), which shows that the design considerations are non-trivial.
> > >
> > > 3) "The answer seems to suggest that the empirical posterior samplers can be close to the true posterior samples. I don't believe this is true..."
> > >
> > > - Response: No, we are not suggesting that the approximate posterior *high-dimensional* image samples are necessarily close the true posterior image samples. Rather, we are suggesting that the empirical quantile of the approximate posterior FRIQ *scalar* samples is nearly as good as the quantile estimate produced by a trained predictor. Thus, our intuition from Sec 3.2 may work well even in the non-ideal situation.

---

### Official Review · Reviewer_K6sy · 2024-11-03

**Soundness:** 2
**Presentation:** 1
**Contribution:** 3
**Rating:** 6
**Confidence:** 3

**Summary:**

This work considers the problem of estimating an $\alpha$-quantile interval on the full reference image quality (FRIQ) metrics in the context of imaging inverse problem. Here FRIQ metrics refers to metrics like PSNR, SSIM, and LIPIPS that compares the reconstructed image with the ground-truth image. The main novelty of the work is the proposal of a method that can estimate FRIQ's interval **without** accessing the ground-truth image, by combining posterior sampling and conformal prediction. The procedure of the proposed method can be roughly schemed as follows:

1. Use an off-the-shelf (approximate) posterior sampling algorithm to reconstruct an single estimate and set of posterior samples.
2. Compute the estimated FRIQ metric between the reconstruction and posterior samples (not using the testing groundtruth image).
2. Given a set of calibrate pairs of the groundtruths and measurements other than the testing groundtruth, perform conformal quantile regression to estimate the bound.

Experiments on image denoising (dataset=FFHQ) and accelerated multicoil MRI (dataset=fastMRI) tasks were conducted to validate the proposed method.

**Strengths:**

1. The considered problem is novel and of sufficient interest to the computational imaging community.
2. The application of conformal prediction to imaging is also relatively new.
3. The current manuscript properly discusses the limitations of the proposed method.

**Weaknesses:**

1. The clarity of the method section needs to be improved.
2. The mathematical notations are abused, which hinders first-time readers to quickly understand the method.
3. The above two points jointly affect proper interpretation of the experimental results.

**Questions:**

1. The definition of adaptiveness seems slightly confusing. From line 216-219, the adaptive methods appear not to depend on the test realization ($\hat{x}_0$) as well. Furthermore, the explanation in line 231-232 is quite vague.
2. The text between line 176-214 appears to focus on the construction of FRIQ samples for the test image, while the reminder of the subsection describes how to apply this construction to all the posterior samples and then conduct CP. The authors are advised to give titles to these texts for clarity.
3. [Section 3.2] It is clear that the non-adaptive method produces just one interval C after calibration. However, the adaptive method seems to produce multiple intervals C's (manifest in $\beta_i$) for each $z_i$. How can eq. 11 be obtained given multiple bounds?
4. [Section 3.3] It appears that the quantile prediction network requires $c$ number of $z_i$ as input. Another limitation of the proposed method is that the network needs to be retrained if $c$ is changed.
5. [Section 3.3] Eq 14 and 15 are not explained. Again, how eq. 16 is obtained remains unclear.
6. [Fig. 1] An addition to figure 1 that illustrates the calibration step is highly recommended.
7. [Fig. 2] Due to the lack of clarity in the method section, it is hard to interpret the left four plots in figure 2, as well as figure 5. Especially, the relationship between $\beta$ and $z$.
8. Can the authors explain what a single Monte-Carlo trial contain? Also, T is not properly defined.
8. Many symbols are used before proper definition or (potentially) doubly defined
    - $U_0$ was used before proper definition.
    - Eq. (2) was not explained.
    - the regularization weight $\lambda$ has been used to denote the empirical miscoverage earlier.

Overall, the current manuscript needs some significant improvement on its clarity. However, I still think the considered problem is interesting, and would look forward to the authors' response.

---

> ### Author Response · Authors · 2024-11-22
>
> We thank the reviewer for their time and feedback.
> We appreciate that the reviewer recognizes the novelty of our method and the significance of bounding the FRIQ.
> Based on your feedback, we made several key modifications and submitted a revised version of our paper.
> The key changes include: 1) simplified notation in Secs. 2 and 3, and 2) the addition of a distribution-shift sensitivity study in App. D.
> All revisions are in colored text.
> Below we address your questions and concerns.
>
> ---
>
> **Weaknesses**
>
> 1) The clarify of the methods section needs to be improved.
>
> - Response: Thanks for your feedback. We've completely rewritten the methods section to make it more clear.
>
> 2) The mathematical notations are abused, which hinders first-time readers.
>
> - Response: Thanks for your feedback. We've revised our notation and we've been careful to ensure that it is not abused.
>
> 3) The above jointly affect interpretation of the experimental results.
>
> - Response: Thanks for the feedback. We believe that our revisions now allow proper interpretation of the experimental results.
>
> ---
>
> **Questions**
>
> 1) The definition of adaptiveness is confusing.
>
> - Response: Thanks for the feedback. By ``adaptive'' we mean that the bounds depend on the measurements $y_0$ and reconstruction $\hat{x}_0$ (See Revision Lines 164-165, 235-237). The revision makes it clear that both the quantile and regression bounds adapt to the measurements $y_0$ through their effect on the image recovery $\\widehat{x}_0$ and the posterior image samples $\\{\\widetilde{x}\_0^{(j)}\\}\_{j=1}^c$, which in turn affect the posterior FRIQs $\\{\\widetilde{z}\_0^{(j)}\\}\_{j=1}^c$, which finally affect the bounds through $\\widehat{z}_0$. Likewise, $\hat{x}_0$ affects the posterior FRIQs $\\{\\widetilde{z}\_0^{(j)}\\}\_{j=1}^c$ and ultimately $\\widehat{z}_0$.
>
> 2) In Original Sec. 3.2, part of the section appears to focus on one topic while the rest focuses on another topic.
>
> - Response: Thanks for your feedback. In the revision, we split that section into two (now Secs. 3.2 and 3.3) to highlight the difference in subject matter.
>
> 3) (Original Section 3.2, Revised Section 3.3) Why does the adaptive method produce multiple bounds per sample?
>
> - Response: We apologize for the confusion. The adaptive method produces only a single bound for each sample, and we have rewritten the text to clarify this fact. For the test sample $z_0$, it provides the bound $\\beta(\\widehat{z}\_0, \\widehat{\\lambda}(d_{\sf cal}))$. For each calibration sample $i\in\{1,\dots,n\}$, it provides the bound $\\beta(\\widehat{z}\_i, \\widehat{\\lambda}(d_{\sf cal}))$.
>
> 4) (Original Section 3.3, Revised Section 3.4) It appears that the quantile prediction network requires $c$ number of $z_i$ as input. Another limitation of the proposed method is that the network needs to be retrained if $c$ is changed.
>
> - Response: To be clear, the quantile prediction network takes in $c$ posterior FRIQ samples $\\{\\widetilde{z}\_i^{(j)}\\}\_{j=1}^{c}$, not $c$ true FRIQs $z_i$. It's true that this network would need to be retrained if $c$ was changed, but we don't see this as a major limitation, since $c$ is a design parameter that would be chosen once and then fixed.
>
> 5) (Original Section 3.3, Revised Section 3.4) Eq (14) and (15) are not explained. And how Eq (16) can be obtained with multiple bounds is unclear.
>
> - Response: For the revision, we moved (14) and (15) to the beginning of Sec. 3 and we put much more effort into explaining them.
>  The revision also clarifies that the learned method produces only a single bound for each sample (just like the other adaptive method).
>
> 6) (Fig. 1) An addition to Fig. 1. showing the calibration step is highly recommended.
>
> - Response: We modified Fig. 1 to better indicate the contents of the calibration set. As for the calibration "step" (i.e., computing $\\widehat{\\lambda}(d_{\sf cal})$ according to Eq. 3), this is the main task of the Conformal Prediction block of Fig. 1.  We use the bisection algorithm for this step, and we're not sure how to easily diagram this algorithm in Fig. 1.  But the bisection algorithm is well-known and we believe that most readers will be comfortable with it.

---

> ### Author Response · Authors · 2024-11-22
>
> 7) (Fig. 2) Due to lack of clarity in the methods section, both Fig. 5 and the left plots in Fig. 2 are hard to interpret, especially the relationship between $\beta$ and $z$.
>
> - Response: We revised the captions of those figures to better explain the contents. Note that, in Section 4 (which contains the numerical experiments), we have many test samples and so we denote them using the subscript "$(\\cdot)\_k$" rather than the subscript "$(\\cdot)\_0$". The $k$th dot in the scatter plot reports the true FRIQ $z_k$ as the horizontal-axis location and the conformal bound $\\beta(\\widehat{z}\_k,\hat{\lambda}(d_{\sf cal}))$ as the vertical-axis location. Meanwhile, the black diagonal line has a slope of 1 and passes through the origin.  So if the $k$th dot falls to the northwest of that line, we know that $\\beta(\\widehat{z}\_k,\hat{\lambda}(d_{\sf cal}))>z_k$. The empirical miscoverage rate $\\alpha$ can then be assessed by measuring the ratio of dots falling on one side of the line versus the other.
>
> 8) What is contained in a single Monte Carlo trial?  Also, $T$ is not property defined.
>
> - Response: Sorry for the confusion. In the revision, we clarify what happens in the $t$-th single Monte Carlo trial and we explain that $T$ is the number of Monte Carlo trials. In each single Monte Carlo trial, we i) randomly draw new calibration and test sets from the available validation data, ii) calibrate the bounding parameter $\lambda$ using the calibration set, and iii) compute the bound $\\beta(\\widehat{z}\_k,\hat{\lambda}(d_{\sf cal}))$ on each sample $k$ of the test set. In each trial, we use 70\% of the validation data for the calibration set we use the remaining 30\% for the test set.
>
> 9) The symbol $U_0$ was used without proper definition, Eq. (2) was not explained, and $\lambda$ was used as both the bound parameter and the regularization weight.
>
> - Response: Sorry for the confusion. In the revision, we've tried to define all quantities before using them and avoid double meanings. We've also added additional explanation around Eq. (2). We believe that our revision is much easier to understand, but we welcome any additional suggestions for improvement.

---

> > ### Comment · Reviewer_K6sy · 2024-11-25
> >
> > I would like to thank the authors for revising the manuscript. However, I may not agree with the authors on point 4 because users could want to choose different values of $c$. I guess that more samples would lead to a more accurate estimate of the bound, but at the cost of a longer runtime for drawing samples. For instance, drawing 100 samples for certain computational imaging tasks can take hours. Hence, users may want to play with this hyperparameter $c$ to ensure a descent trade off between accuracy and time.

---

> > > ### Author Response · Authors · 2024-11-25
> > >
> > > Indeed, users may want to choose a different value of $c$ to trade-off bounding performance with inference time, and the quantile bound has an advantage over the regression bound in that the former does not require re-training a (scalar) predictor.  For our experiments, we used a polynomial-based predictor that trained very quickly, but more generally one might use a predictor that takes a long time to train.
> > >
> > > However, we note that one would not need to regenerate samples for every value of $c$. One could  pre-compute and save the posterior FRIQs $\\{\\widetilde{z}\_i^{(j)}\\}\_{j=1}^c$ for the training samples with a conservatively large value of $c$, e.g. $c=100$. By loading these saved values and only using a subset, one could train $f$ for any value of $c\leq100$ without needing to regenerate posterior samples. This improves the efficiency of training for different values of $c$ significantly.

---

### Official Review · Reviewer_1Tr7 · 2024-11-04

**Soundness:** 3
**Presentation:** 3
**Contribution:** 3
**Rating:** 6
**Confidence:** 4

**Summary:**

This paper proposes conformal bounds on full-reference image quality metrics without access to the true image, which can be utilized in safety-critical applications, such as calculating the extent of possible in accelerated MR imaging given a predefined error tolerance.

**Strengths:**

**Novelty and potential real-life application**

* The paper utilizes conformal prediction (CP) , which is well-suited for generating uncertainty bounds, to calculate bounds on full-reference image quality metrics. This can be particularly useful in medical imaging, especially in scenarios like the multi-round MRI acquisition described in lines $463-464$.
* Multiple bounds, including an adaptive bound and its improved version, are provided, and numerical comparisons between each bound are made. Theoretical claims are supported by the detailed experiments provided in appendices.

**Weaknesses:**

**Most of the weaknesses are identified and discussed in the Limitations paragraph (Line $509$)**

* As stated, the method may not work if the calibration and test data distributions are significantly different. I acknowledge it requires more work to make the method more robust to distribution shift, but a simple experiment demonstrating the sensitivity to the such shifts can be added.
* The performance difference between the adaptive bound and the improved adaptive bound appears incremental in Figure 4 and 8.

**Questions:**

* How do the authors explain the insensitivity of Mean Conformal Bound to the Number of Image Samples $c$, which may not be intuitive?
* In a resource-constrained setting, could calculating adaptive bounds be challenging, especially for a real-time implementation?

**Suggestions**

* Adding a numerical experiment to demonstrate the effect of a simple distribution shift between the test and calibration data would be interesting and increase the paper's impact.
* As a theoretical limit, the acceleration factor which can be resolved using parallel imaging (PI) is constrained by the number of receiver coils; thus, it is better to state line $413-414$ as "For $R>1$, the inverse problem _might_ become ill-posed." It is guaranteed to become ill-posed for single coil imaging, but for PI, coil sensitivity maps affect the linear system as well.

---

> ### Author Response · Authors · 2024-11-22
>
> We thank the reviewer for their time and feedback.
> We are glad the reviewer appreciates the novelty and real-world potential of our approach and believes the work is well-grounded in theory and supported by the experimentation.
> Based on your feedback, we made several key modifications and submitted a revised version of our paper.
> The key changes include: 1) simplified notation in Secs. 2 and 3, and 2) the addition of a distribution-shift sensitivity study in App. D.
> All revisions are in colored text.
> Below we address your questions and concerns.
>
> ---
>
> **Weaknesses**
>
> 1) A simple experiment studying the sensitivity to distribution shifts could be added.
> - Response: Thank you for this suggestion. We included such a study in App. D of the revision, where we calibrated on the center slices of 3D MRI volumes and tested on slices taken from increasingly larger distances from the center. The study shows, both qualitatively and quantitatively, that certain metrics (e.g., PSNR and SSIM) are relatively robust to small shifts while others are less so.
>
> 2) Limited performance improvements between the adaptive bound and the learned regression bound
> - Response: Yes, there is a limited performance gap, but we're not sure that this is a weakness. Rather, we believe that it attests to the accuracy of the design intuitions provided in Sec. 3.2, which establish that the ideal conformal bound can be computed using an empirical quantile of an infinite number of perfect-posterior samples. In particular, our results suggest that relaxing the ideal scheme to use a finite number of approximate posterior samples is nearly as good as training a bound-predictor from scratch.
>
> ---
>
> **Questions**
>
> 1) Why is the Mean Conformal Bound insensitive to the number of posterior samples $c$?
> - Response: We conjecture that an insensitivity to $c$ is likely to arise whenever the posterior samples $\\{\\widetilde{z}\_{i}^{(j)}\\}\_{j=1}^{c}$ have a small variance. This seems to be the case for our experiments, but it may not happen in general.
>
> 2) Could computing adaptive bounds be challenging for real-time applications under resource-limited conditions?
> - Response: Perhaps, although we believe that computation issues can be mitigate by good design choices.  The most computationally demanding step of our approach is generating posterior samples, for which the required effort depends on the number of samples $c$ and the computation per sample.  Our experiments suggest that a small $c$ suffices for tight bounds, at least in some applications.  And while diffusion samplers may be slow, there is growing evidence that well-designed GANs offer competitive performance with much less compute.
>
> ---
>
> **Suggestions**
>
> 1) Add a numerical experiment to demonstrate the effect of a distribution shift.
> - Response: Thank you for your suggestion. We have included such an experiment in App. D of the revision.
>
> 2) Clarify that acceleration in parallel MRI **might** lead to ill-posedness, but is not guaranteed to.
> - Response: Thank you for your suggestion. We have added this clarification.

---

> > ### Comment · Reviewer_1Tr7 · 2024-12-01
> >
> > I want to thank the authors for their thoughtful responses to my comments and suggestions. The addition of the sensitivity study in App. D and the clarifications on acceleration in parallel imaging demonstrate a genuine effort to address the concerns raised. These revisions have increased my confidence in the soundness and applicability of the work, and I am raising my confidence score from 3 to 4. I feel that the core contribution remains consistent with my original evaluation. Thus, I am maintaining the original score of 6.

---

### Official Review · Reviewer_8V4H · 2024-11-07

**Soundness:** 3
**Presentation:** 2
**Contribution:** 3
**Rating:** 5
**Confidence:** 3

**Summary:**

This paper presents a method for constructing prediction intervals for Full-Reference Image Quality (FRIQ) metrics in imaging inverse problems based on conformal prediction. By leveraging a calibration set and empirical FRIQ samples generated from posterior reconstructions, the approach provides intervals that offer coverage guarantees on the true quality metric for new test images. Specifically, the method selects a parameter $\lambda$ to balance the interval width and the desired error rate, ensuring that the intervals meet a specified confidence level, typically $1−\alpha$. Numerical evaluations on natural image denoising and accelerated MRI reconstruction demonstrate the potential practicality of this approach.

**Strengths:**

1, The research question is significant from various perspectives, especially for trustworthy machine learning, as it addresses the need for reliable uncertainty quantification in image reconstruction tasks.

2, Another strength of this paper’s novelty is its integration of conformal prediction with approximate posterior sampling to construct statistically rigorous bounds on FRIQ metrics for imaging inverse problems, offering guaranteed coverage with a user-specified error probability. This approach provides robust uncertainty quantification in complex imaging tasks where data distributions are unknown.

**Weaknesses:**

1, The paper in its current form lacks a more comprehensive review of other uncertainty quantification methods, such as Bayesian approaches (e.g., Monte Carlo dropout), which are widely used in imaging reconstruction. Including these comparisons would better highlight the proposed method’s advantages in terms of coverage guarantees and reliability.

2, Similarly, the paper also lacks numerical comparisons with other uncertainty quantification methods beyond conformal prediction. Including these would provide a clearer assessment of the proposed method’s effectiveness relative to established approaches.

3, The proposed approach seems to rely on an exchangeability assumption between calibration and test data, which may not hold in real-world imaging due to distribution shifts. This could limit the method's robustness, especially with diverse or evolving datasets, like those in medical imaging across different populations or devices.

4, The numerical results in Sections 4.1 and 4.2 show no clear performance gain between adaptive bound estimation and its improved learning variants, which may weaken the methodological contribution of the proposed approach.

**Questions:**

1, Is the choice of calibration $d_{cal}$ critical to the success of this method? How would the intervals behave if $d_{cal}$ were generated using a different model than the one used for predictions? Given that $d_{cal}$  is assumed to represent the true distribution, how sensitive is the method to shift between the calibration distribution and the test distribution?

2, How does the number of FRIQ samples influence the quality of the prediction interval? Is there a minimum number of samples required for reliable interval construction? Since the empirical miscoverage rate uses an indicator function, does it introduce any quantization errors at the same time?

3, The reconstruction results in Figures 2, 3, and 4 rely on MMSE approximations, which result in poorer perceptual quality. Could the authors instead use a single DDRM sample for reconstruction and test the coverage, potentially improving perceptual quality while evaluating the coverage of this approach?

4, Consistency in notation across sections, particularly between Sec. 2 and Sec. 3, would improve readability and understanding. Making sure that the terms introduced in the CP background (Sec. 2) align directly with their usage in the adaptation in Sec. 3 could bridge any gaps for readers. This way, each section builds on the previous one without introducing confusion.

---

> ### Author Response · Authors · 2024-11-22
>
> We thank the reviewer for their time and feedback.
> We appreciate that the reviewer acknowledges the significance of our approach in providing uncertainty quantification to inverse imaging problems and recognizes the novelty of our incorporation of conformal prediction.
> Based on your feedback, we made several key modifications and submitted a revised version of our paper.
> The key changes include: 1) simplified notation in Secs. 2 and 3, and 2) the addition of a distribution-shift sensitivity study in App. D.
> All revisions are in colored text.
> Below we address your questions and concerns.
>
> **Weaknesses**
>
> ---
>
> 1) Lack of a comprehensive review of other uncertainty quantification methods like Bayesian/dropout methods, especially with regards to coverage guarantees and reliability.
>
> - Response: In the revision (Lines 64-89), we include a review of existing uncertainty quantification (UQ) methods, including Bayesian/dropout methods. But in the end we emphasize the following two key points:
>     a) Apart from those methods based on conformal prediction, existing UQ methods (e.g., Bayesian/dropout) provide no coverage guarantees whatsoever.
>     b) No existing UQ methods target FRIQ.
>
> 2) Lack of numerical comparison to other uncertainty quantification methods.
>
> - Response: Since we are unaware of any existing UQ methods that target FRIQ (and certainly none that provide guaranteed bounds on FRIQ) we are unsure of what to numerically compare against.  But if the reviewer knows of some, we'd be happy to discuss them and compare against them.
>
> 3) The exchangeability requirement may limit robustness.
>
> - Response: Indeed, we acknowledged the limitations of the exchangeability assumption in the original submission (Lines 509-512). A limitation of some form is to be expected, since every statistical guarantee relies on a particular set of assumptions. To better understand the robustness of our method, we added a numerical study that analyzes the sensitivity of our method to distribution shifts between calibration and test data (Revision App D). In any case, we believe that our paper provides a foundation for future work on safe and reliable imaging, as progress continues to be made in robust conformal prediction.
>
> 4) Limited performance improvement between the quantile-based bound and learned regression bound.
>
> - Response: Yes, there is a limited performance gap, but we're not sure that this is a weakness. Rather, we believe that it attests to the accuracy of the design intuitions provided in Sec. 3.2, which establish that the ideal conformal bound can be computed using an empirical quantile of an infinite number of perfect-posterior samples. In particular, our results suggest that relaxing the ideal scheme to use a finite number of approximate posterior samples is nearly as good as training a bound-predictor from scratch.
>
> ---
>
> **Questions**
>
> 1) How does the choice of $d_{\sf cal}$ affect the method? How sensitive is the method to a distribution shift between the calibration and test set?
> - Response: For the coverage guarantees to hold, our method requires only that the calibration samples in $d_{\sf cal}$ are exchangeable with the test sample. To better understand the effect of distribution shift between the calibration and test data, the revision includes a numerical study in App. D. That study provides evidence that certain metrics (e.g., PSNR and SSIM) are relatively robust to small distribution shifts.
>
> 2) How does the number of FRIQ samples influence the quality and reliability of the prediction interval?  Does the use of an indicator function for empirical miscoverage induce quantization errors?
> - Response: If the reviewer is asking about the number of calibration samples $n$ in $d_{\sf cal}$, the coverage guarantee will hold for any $n$, but the bounds will become more conservative as $n$ decreases (see Eq. 3). If the reviewer is asking about $c$, the number of FRIQ estimates $\\{\widetilde{z}\_{i}^{(j)}\\}\_{j=1}^c$ used to compute the quantile and regression bounds, the coverage guarantee holds for any $c\geq 0$, but the bounds tend to become more conservative as $c$ decreases, as seen in Fig. 4. Empirical miscoverage is, by definition, quantized. The quantization becomes more coarse as $n$ decreases, and in response the bounds become more conservative (see Eq. 3). But the bounds remain valid for any $n$.
>
> 3) Figures 2,3,4 rely on MMSE approximations. Could the authors instead use a single DDRM sample for the reconstruction?
> - Response: We did use a single DDRM sample for the reconstructions in Figs. 2,3,4, as reported in Original Line 354-355.  The revision further highlights this fact.
>
> 4) Consistency in notation between Secs. 2 and 3.
> - Response: Thank you for this suggestion. We've significantly revised both Sec.s 2 and 3 to make the notation simpler and more consistent.

---

### Meta-Review · Area_Chair_VKJ6 · 2024-12-16

**Metareview:**

This paper proposes a method to generate reliable bounds on full-reference image quality (FRIQ) metrics in imaging inverse problems.  The method combines conformal prediction with approximate posterior sampling to construct bounds on FRIQ metrics (e.g., PSNR, SSIM, LPIPS) that are guaranteed to hold up to a user-specified error probability.  The authors demonstrated their approach on image denoising and accelerated magnetic resonance imaging (MRI) problems.

The strengths of the paper include addressing an important problem with a well-motivated solution, proposing a sound method with guaranteed bounds, and providing experimental results to demonstrate the effectiveness of the method.

The weaknesses include limited technical novelty, lack of clarity, and limited experimental validation.  The paper could be improved by including a comprehensive review of other uncertainty quantification methods, a discussion of the limitations of the proposed method, and an analysis of the sensitivity of the method to distribution shifts.

**Additional Comments On Reviewer Discussion:**

During the rebuttal period, the reviewers raised concerns about the clarity, novelty, and experimental validation of the paper.  The authors responded to these concerns by simplifying the notation, adding a distribution-shift sensitivity study, and providing additional explanations.  However, the reviewers were not fully satisfied with the authors' response and the concerns about the novelty of the paper remained.  These unresolved concerns played a significant role in the final decision to reject the paper.

---

### Decision · Program_Chairs · 2025-01-22

Reject